# Earth-abundant Ni-Zn nanocrystals for efficient alkyne semihydrogenation catalysis

Jasper Clarysse[1,3], Jordan De Jesus Silva[2,3], Yunhua Xing [1],
Seraphine B. X. Y. Zhang[2], Scott R. Docherty[2], Nuri Yazdani [1],
Maksym Yarema [1], Christophe Copéret [2,4] ✉ & Vanessa Wood [1,4] ✉

The development of catalysts that are based on earth-abundant metals remains a grand challenge. Alloy nanocrystals (NCs) form an emerging class of heterogeneous catalysts, offering the promise of small, uniform catalysts with composition-control. Here, we report the synthesis of small Ni and bimetallic Ni-X (X= Zn, Ga, In) NCs for alkyne semihydrogenation catalysis. We show that Ni$_3$Zn NCs are particularly reactive and selective under mild reaction conditions and at low loadings. While bimetallic NCs are all more selective than pure Ni NCs, Ni-Zn NCs also maintain excellent reactivity compared to Ni-Ga and Ni-In alloys. Ab-initio calculations can explain the differences in reactivity, indicating that, unlike Ga and In, Zn atoms interact with the substrates. We further show that Ni$_3$Zn NCs are robust and tolerate a broad range of substrates, which may be linked to the favorable amine-terminated surface.

The selective semihydrogenation of alkynes is one of the most fundamental organic transformations with applications in both the fine chemical industry (e.g., synthesis of vitamins and natural products)[1] as well as in the polymer industry (e.g., selective removal of acetylene to purify ethylene streams)[2,3]. Since its discovery in the early 1950s, the Lindlar catalyst (Pd/Pb(OAc)$_2$ on CaCO$_3$)[4] has been the standard catalyst for *Z*-selective alkyne semihydrogenation. The drawbacks of this catalyst include the scarcity and increasing prices of Pd, the inherent toxicity of Pb, its limited selectivity, and its relatively narrow substrate scope. Therefore, substantial effort has been directed at developing alternative homogeneous and heterogeneous alkyne semihydrogenation catalysts, based on earth-abundant[5] and less environmentally impactful metals[6–8]. For homogeneous catalysis, this includes non-precious metal complexes, e.g. Cu, Ni, Fe and Mn, including N-heterocyclic carbene (NHC)[9–13], pincer[14–19], bipyridine[20], metalloligand[21,22] and phosphine[23–25] complexes. For heterogeneous alkyne semihydrogenation catalysts, coinage metals (Cu, Ag, Au) have been shown to display high selectivity in alkyne semihydrogenation due to the weaker coordination of the alkenes towards such metals[26–28], but typically require harsher reaction conditions (high temperatures and/or H$_2$-pressures) to overcome their lower catalytic activity. Group 10 metals (Ni, Pd, Pt) display good

hydrogenation reactivity at the expense of selectivity, resulting in overhydrogenation. Several approaches have been explored to improve their selectivity such as metal-alloying[2,28–30], surface capping with ligands[4,31–35], single atom catalysts[6], and partial surface oxidation[36–38].

In this context, nanocrystals (NCs) can offer several advantages: they can be prepared as small and uniform particles that are size- and composition-tunable with changeable surface chemistry[39,40]. This tunability enables NCs to be used for the systematic study of catalytic reactions and for the optimization of catalyst composition, which is key to obtain the desired balance between activity and selectivity[26,41,42]. Wet-chemical colloidal synthesis approaches allow the production of NCs with excellent size and composition control, yielding stable and easily processable dispersions[40,43,44], but the colloidal synthesis of metal-alloy NCs is particularly challenging as metals have different reduction potentials, oxophilicity, and chemical characteristics. Alkyne semihydrogenation catalyst formulations synthesized via colloidal synthesis have been demonstrated[30,43,45–47], for instance of earth-abundant Ni-based NCs, diluted with a second element such as boron[34], phosphorous[48], or alloyed with a second metal[45,49–52]. Yet, synthetic progress is needed to achieve more size- and compositionally-uniform NCs with pristine, non-oxidized surfaces,

[1]ETH Zurich, Institute for Electronics, Department of Information Technology and Electrical Engineering, Zurich, Switzerland. [2]ETH Zurich, Department of Chemistry and Applied Biosciences, Zürich, Switzerland. [3]These authors contributed equally: Jasper Clarysse, Jordan De Jesus Silva. [4]These authors jointly supervised this work: Christophe Copéret, Vanessa Wood. ✉e-mail: ccoperet@ethz.ch; vwood@ethz.ch

while avoiding capping ligands such as phosphines, thiols, or carboxylic acids, which reduce or diminish the catalytic activity of NCs[39].

Recently, amalgamation seeded growth, which involves the colloidal synthesis of alloy NCs via the non-epitaxial nucleation of low-melting point metals on pre-synthesized seeds at elevated temperatures, was developed[53]. With amalgamation seeded growth, the amount of an alloying element can be quantitatively introduced such that predictive composition control is possible. This synthetic approach is thus particularly powerful to produce a range of NCs suitable to investigate the role of catalyst composition.

Here, we focus on earth-abundant Ni-based catalysts for alkyne semihydrogenation and employ the amalgamation method to achieve bimetallic Ni-X NC catalysts, with the aim to identify Ni-alloys that are both reactive and selective. As alloying metals, we explore Ga, In, and Zn, as these are selectivity promoters for catalysts in various hydrogenation reactions such as $CO_2$-hydrogenation[54,55], alkyne semihydrogenation[45], the chemoselective hydrogenation of α,β-unsaturated carbonyl compounds[56], electrocatalytic transformations[57,58] and $H_2$-based deoxygenation reactions[59]. Via amalgamation seeded growth, we achieve small and highly monodisperse NC catalysts with a range of compositions, systematically varying Ni-content and using the different alloying elements (X = Ga, In, Zn). We test the reactivity and selectivity of the different Ni-X compositions and rationalize the trends using density functional theory (DFT) calculations. The top performing composition, $Ni_3Zn$, offers both high conversion and selectivity, so we then focus on understanding its performance and functional group tolerance across a broad substrate scope.

## Results

### Synthesis and characterization of nanocrystal catalysts

Bimetallic Ni-Ga NCs have previously been achieved using amalgamation seeded growth[53]; however, the Ni seeds in this first study were relatively large NCs (>10 nm), which is not ideal for semihydrogenation catalysis[26]. Furthermore, monodispersity was achieved with trioctylphosphine ligands, which are typical catalyst poisons that can inhibit or surpress activity[60].

We therefore develop a facile synthesis procedure to obtain small NCs without oxidation and with suitable ligand chemistry. First, we achieve small Ni NCs using a hot-injection procedure involving tert-butylamine borane (tBAB), a reducing agent that is highly soluble in organic solvents[61]. Bimetallic Ni-based NCs have already been reported from the co-reduction of $Ni(acac)_2$ and a second metal acetate or acetylacetonate precursor in oleylamine and oleic acid with tBAB, yet these NCs are partially oxidized, reducing catalytic activity[45]. Since amine ligands themselves have been shown not to poison semihydrogenation catalysts[39], even improving selectivity, as reported for octylamine coordinated on Pt and $Pt_3Co$ NCs[33], we employ oleylamine ligands solely and find that this results in non-oxidized, small and monodisperse ($3.2 \pm 0.5$ nm) Ni NCs. Here, we use oleylamine instead of octylamine since the former has a higher boiling point, allowing subsequent alloying of the colloidal NCs at elevated temperatures.

For amalgamation, instead of previously used metal acetate and acetylacetonate precursors, we use metal-amide and -silylamide precursors, which possess reactive metal-nitrogen bonds that are thermally cleavable at temperatures attainable in colloidal synthesis procedures[62], and which do not oxidize Ga, In, or Zn upon decomposition.

To prevent oxidation upon post-synthetic purification, we employ anhydrous methyl acetate as an aprotic and weakly coordinating anti-solvent instead of more common protic anti-solvents found in colloidal synthetic purification protocols such as ethanol.

To study the influence of the composition of the NCs on alkyne semihydrogenation catalysis performance, we use this approach to then synthesize Ni-based alloy NCs, targeting the compositions $Ni_8Zn$, $Ni_3Zn$, and NiZn as well as $Ni_3Ga$ and $Ni_3In$ to study the influence of the alloy composition and stoichiometry. The achieved NCs compositions are measured using Scanning Electron Microscopy Energy Dispersive X-ray (SEM EDX) (Supplementary Table 1) and, within the experimental error, match the expected compositions. For all compositions, highly monodisperse fcc-structured NCs which are small in size (respectively $3.2 \pm 0.4$ nm, $3.4 \pm 0.4$ nm, $3.6 \pm 0.4$ nm $4.0 \pm 0.5$ nm and $3.9 \pm 0.4$ nm for Ni, $Ni_8Zn$, $Ni_3Zn$, NiZn, $Ni_3Ga$ and $Ni_3In$) are achieved (Fig. 1b), as observed from Transmission Electron Microscopy (TEM) images and

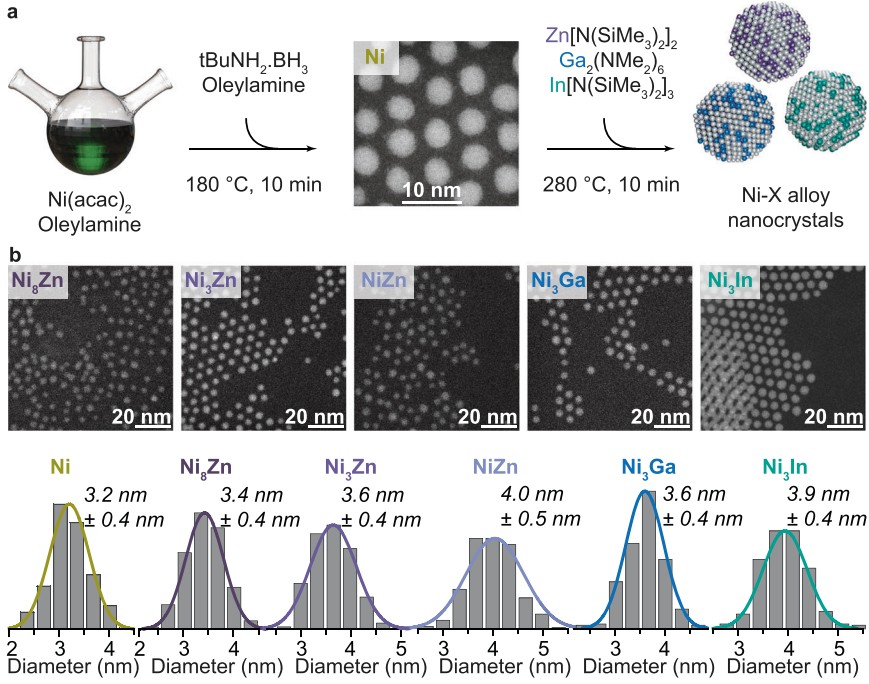

**Fig. 1 | Synthesis of earth-abundant Ni-X (X = Ga, In, Zn) bimetallic nanocrystals. a** Reaction scheme for the synthesis of Ni NCs and bimetallic $Ni_8Zn$, $Ni_3Zn$, NiZn, $Ni_3Ga$, and $Ni_3In$ NCs. **b** Scanning transmission electron microscopy (STEM) images of the NCs along with corresponding size distribution histograms.

X-ray Diffraction (XRD) patterns (Supplementary Figs. 1 and 2). The XRD patterns feature broadened reflections due to the small size and polycrystalline nature of the NCs. A high-resolution TEM image, Supplementary Fig. 3, of a $Ni_3Zn$ nanocrystal indicates its fcc structure and polycrystaline nature. Fourier-Transform Infrared Spectroscopy (FTIR) measurements of $Ni_3Zn$ NCs, Supplementary Fig. 4, confirm the presence of oleylamine ligands on the NCs. Comparing the X-ray Absorption Near Edge Structure (XANES) of X-ray Absorption Spectroscopy (XAS) measurements of $Ni_3Zn$ and Ni NCs, Supplementary Fig. 5, reveals the zerovalency of the elements in the NCs and charge transfer occuring in the NiZn alloy from Zn to Ni. The reduced state of the elements within all as-synthesized NCs is also observed from X-ray Photoelectron Spectroscopy (XPS) (Supplementary Figs. 6 and 7).

## Semihydrogenation of 1-Phenyl-1-Propyne with Ni-X NCs

The as-synthesized NCs are tested for their performance as alkyne semihydrogenation catalysts (Supplementary Fig. 8). 1-Phenyl-1-propyne is selected as a prototypical substrate, as it is representative for the large class of internal alkynes containing an aromatic moiety.

To assess the NCs as catalysts (Fig. 2), we perform Gas Chromatography coupled simultaneously to a Flame Ionization Detector and a Mass Spectrometer (GC-FID-MS) after each experiment, using $n$-tridecane as an internal standard. We consider conversion, defined as the amount of 1-phenyl-1-propyne that has reacted for the set time of the reaction, and selectivity, defined as the percentage of $cis$-$β$-methyl styrene ($Z$-2H) in the total amount of products from the reaction. We also account for the carbon balance of each experiment, comparing the total number of carbon atoms from the remaining reactant and the products formed after the reaction with GC-FID-MS to the total number of carbon atoms in the initial reactant substrate, which is also analyzed with GC-FID-MS. When the carbon balance is not 100 %, we attribute this to side-reactions, such as cyclotrimerization or oligomerization reactions. The reproducibility of the measurements is high with a ±3.5% variation in selectivity for 1-phenyl-1-propyne found across different runs (Supplementary Table 2). Although not strictly quantitative, to differentiate among the reactivity of catalysts, particularly those exhibiting full conversion during the reaction time, we also consider the rate of $H_2$-uptake, observed from the $H_2$-uptake curves recorded throughout each reaction.

Since $Ni_3Zn$ is at the center of our composition study, we perform a systematic screening with $Ni_3Zn$ NCs to select reaction parameters

for testing all NCs (Supplementary Table 3, Supplementary Figs. 9 and 10), and find that under reaction conditions of 80 °C and 1 bar $H_2$, full conversion is achieved together with high selectivity (86%) for the semihydrogenated product(s). Notably, increased reaction times (16 h) do not compromise selectivity (Supplementary Fig. 11). In addition, the catalyst loading can be reduced to 0.5 mol%, while still resulting in full conversion after a 10 h reaction time. These are mild conditions for catalysts consisting of earth-abundant metals, which otherwise require high $H_2$-pressures up to 50 bars and elevated temperatures for alkyne semihydrogenation[31,43].

Using the determined conditions (80 °C, 1 bar $H_2$, 0.5 mol% per Ni catalyst loading, 10 h), we compare the catalytic performance of the different compositions (Fig. 2, Supplementary Table 4, and Supplementary Figs. 12–14). Monometallic Ni NCs display 100% conversion of 1-phenyl-1-propyne but are unselective, yielding the overhydrogenated product $n$-propyl benzene. Alloying with Zn maintains conversions of 100% for $Ni_8Zn$ and $Ni_3Zn$ NCs, respectively, while progressively enhancing the selectivity (64% for $Ni_8Zn$ and 85% for $Ni_3Zn$ NCs). Increasing the Zn content to 50% (NiZn), however, decreases the reactivity of the NCs such that after 10 h of reaction, only 43% conversion of 1-phenyl-1-propyne had occurred (this decrease in reactivity for NiZn NCs compared to $Ni_3Zn$ NCs is relatively small considering that the employed hydrogenation conditions, e.g., 1 bar $H_2$, are mild). Our findings that the 3:1 composition provide a suitable balance between selectivity and reactivity are in line with other literature reports (e.g., $Ni_3N$[63], $Ni_3Sn$[45,64], $Ni_3Ge$[65], $Ni_3Ga$[45,64], $Ni_3In$[64], $Pt_3Co$[33]).

$Ni_3Ga$ and $Ni_3In$ NCs are less reactive compared to $Ni_3Zn$ NCs under these reaction conditions, reaching 12% and no conversion after 10 h, respectively. However, these catalysts do show complete conversion and selective semihydrogenation of 1-phenyl-1-propyne under slightly harsher conditions (5 bar $H_2$, 16 h, 80 °C, 0.5 mol% catalyst), Supplementary Fig. 15.

## Origin of reactivity trends

To understand the atomic origins of the trends in performance of Ni, $Ni_3Zn$, $Ni_3Ga$, and $Ni_3In$ NCs, we perform Density Functional Theory (DFT) calculations as described in the Methods[30,33]. First, we generate slabs with surfaces that are representative of the NC surface (Supplementary Fig. 16) by calculating the approximate surface content of the metal species on the NCs from relative sensitivity factors (RSFs) on the measured Ni $2p_{3/2}$ XPS spectra (Supplementary Table 5 and

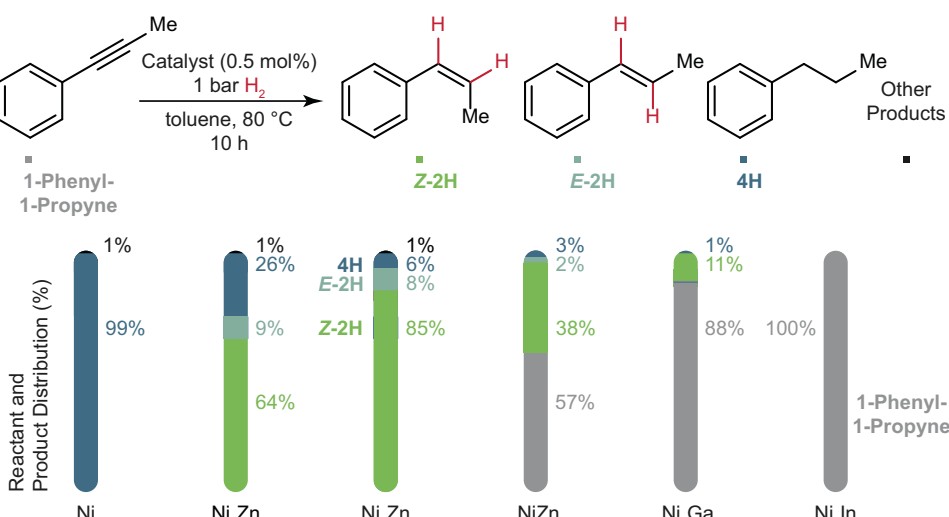

**Fig. 2 | Catalyst composition comparison for 1-phenyl-1-propyne semihydrogenation.** Catalytic reaction scheme and reactant and product distribution for 1-phenyl-1-propyne semihydrogenation. The substrate 1-phenyl-1-propyne is shown in gray, $Z$-2H ($cis$-$β$-methyl styrene) in green, $E$-2H ($trans$-$β$-methyl styrene) in light blue, 4H ($n$-propyl benzene) in dark blue, and other products (based on carbon balances) in black.

Supplementary Figs. 17 and 18). From the XPS spectra, we derive that all Ni₃X NCs show approximately 50% surface content of alloying metal X instead of the 25% expected. This is consistent with Scanning-Tunnelling Electron Microscopy (STEM) Energy Dispersive X-Ray (EDX) elemental mappings, Supplementary Fig. 19, which indicate a Zn-enriched surface on Ni₃Zn NCs. As the Ni₃Zn NCs have an approximately 1:1 surface stoichiometry, they feature the surface composition which was predicted by F. Studt et al. to be highly promising for alkyne semihydrogention catalysis[42]. The fact that some Zn, Ga, and In is drawn to the surface of the alloy NCs might be explained by the post-synthetic purification procedure with methyl acetate and the capping of the surface with oleylamine ligands[49]. The small differences in surface content of Zn, Ga, and In is in agreement with the solubility differences of the alloying elements in face-centered cubic (fcc) Ni, Supplementary Fig. 18. We therefore use model slabs of Ni and Ni-alloys with a stoichiometry of the upper two layers set to a 50:50 ratio (and random mixing) in the DFT-calculations.

It has been previously shown for Pd-Ag[66] and Ni-Ga[45] that alloying has a large effect on the adsorption energies of alkynes and alkenes but negligible effect on activation barriers for the hydrogenation steps. We confirm this for Ni-X with transition state calculations, which show similar energy barriers for Ni, Ni-Ga, Ni-In, and Ni-Zn (Supplementary Fig. 19 and Supplementary Tables 6 and 7). This means that the trends in catalyst activity and selectively can primarily be linked to the binding energies of the substrate and the products to the catalyst surface[42].

DFT-calculated adsorption energies ($E_{ads}$) of alkynes, alkenes, and H on Ni and Ni-X slabs are shown in Fig. 3 (Supplementary Fig. 20, Supplementary Table 8). Calculated $E_{ads}$ for hydrogen are approximately the same for all compositions, but $E_{ads}$ for 1-phenyl-1-propyne decreases from Ni to Ni₃Zn to Ni₃Ga to Ni₃In, which is consistent with the reactivity of the NCs decreasing from Ni to Ni₃Zn to Ni₃Ga to Ni₃In. The absolute value of $E_{ads}$ for 1-phenyl-1-propene is lower for Ni₃Zn, Ni₃Ga, and Ni₃In surfaces compared to Ni, consistent with our experimental finding that all Ni-X alloys show better selectivity than Ni (Supplementary Table 4). The agreement between the trends in DFT-calculated adsorption energies and the experimental results suggest that we can use DFT to study the atomic origins of reactivity and selectivity in our Ni-X NCs.

To eliminate the influence of any functional group, we select ethylene and acetylene as model substrates and examine how they interact with the computed Ni and Ni-X surfaces. We find that the most energetically favorable configuration for acetylene on Ni and Ni-X surfaces is a bridged position within a parallelogram formed by four atoms, Supplementary Table 8. For acetylene adsorption, the parallelogram-site is found to be more favorable compared to a tri-angular site formed by Ni-atoms, which was reported as the catalytic site in case of polyhydride Ni-Ga clusters[21] and Ni₅Ga₃ intermetallic NCs[51] (indicating that the small fcc-structured solid solution alloy NCs reported here are structurally unique materials for catalysis). The Projected Density of States (PDOS) of acetylene on Ni and Ni-X surfaces reveal that the Ni and Zn *d*-orbitals hybridize with the *p*-orbitals of acetylene carbon atoms, while the *d*-orbitals of Ga and In do so to a much smaller extent, Supplementary Fig. 21. Therefore, in Ni-Zn alloys, both Ni and Zn-atoms contribute to the adsorption of alkynes, while for Ni-Ga and Ni-In alloys, the Ni atoms are primarily responsible for the bonding. This is visualized in electron density maps (Supplementary Fig. 22) and explains why Ni-Zn NCs maintain excellent activity compared to Ni-In and Ni-Ga NCs.

For the case of the product molecule (represented by ethylene), we find that Ni surfaces offer two binding configurations with comparable total energy, whereas for Ni-Zn, Ni-Ga, and Ni-In surfaces, only one of these configurations (a triangle-site formed by Ni-atoms[21,51]) is energetically favorable, Supplementary Table 8. As visualized in electron density maps (Supplementary Fig. 23), alloying elements in the slab do not contribute to the adsorption of ethylene. This explains the reduction in the binding strength between alkenes and the alloy NCs compared to Ni NC surfaces, and the fact that Ni-Zn, Ni-Ga, and Ni-In alloys are more selective relative to pure Ni NCs.

## Catalytic performance of Ni₃Zn NCs

Given the superior performance of the Ni₃Zn NC catalysts, we investigate these in more detail. Using the standard conditions (80 °C, 1 bar H₂, 0.5 mol% catalyst loading), we monitor the reaction with regular 2h-sampling (Fig. 4a). The zero-order dependence of reaction rate on alkyne concentration for the first 6–8 hours indicates that the adsorption of alkynes onto the Ni₃Zn NCs is not rate determining under these conditions. The steady increase in conversion to *cis-β*-methylstyrene and slow conversion to the overhydrogenation products (*n*-propyl benzene) demonstrates the preferential interaction of the alkynes over the alkenes with Ni₃Zn NCs, consistent with the adsorption energies calculated via DFT.

Recycle tests reveal that the catalytic performance (reactivity and selectivity) of the Ni₃Zn NCs does not drop significantly over three runs

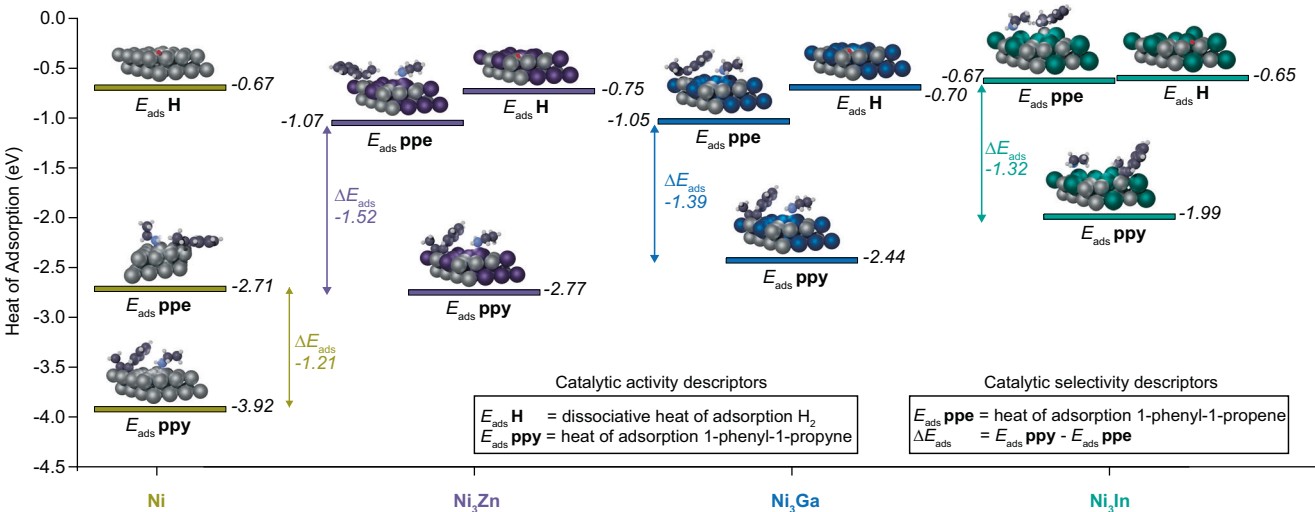

**Fig. 3 | Catalyst composition dependent reactivity trends.** Density functional theory (DFT)-calculated heat of adsorption values for hydrogen, 1-phenyl-1-propyne (ppy), 1-phenyl-1-propene (ppe), on 111-faceted slabs of Ni, Ni₃Zn, Ni₃Ga and Ni₃In with a 1:1 stoichiometric ratio for the outer two atomic layers of the alloys. Ethylamine is co-adsorbed as a surrogate for oleylamine capping ligands.

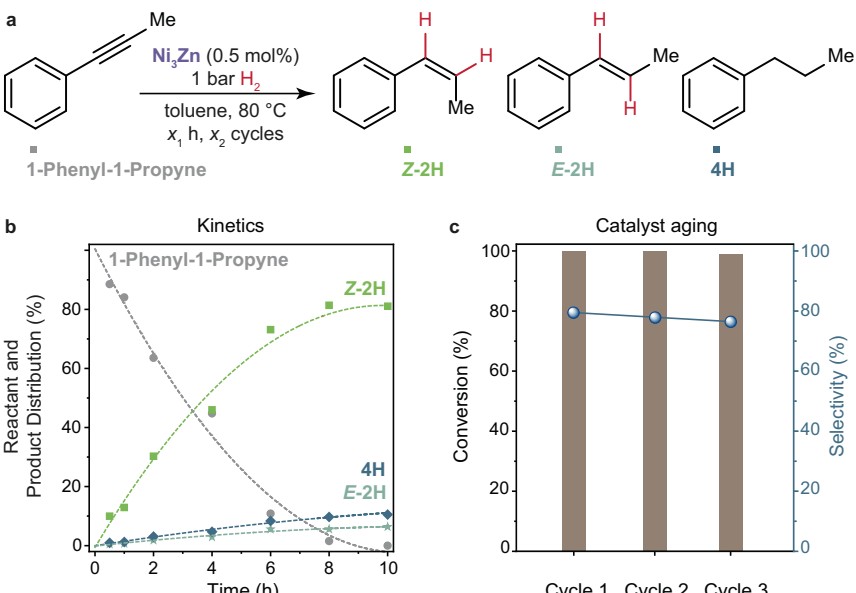

**Fig. 4 | Catalytic performance of Ni₃Zn NCs for 1-phenyl-1-propyne semihydrogenation. a** Reaction scheme for 1-phenyl-1-propyne (1 mmol) hydrogenation. **b** Time-dependent conversion of 1-phenyl-1-propyne using 0.5 mol% Ni₃Zn NCs and

1 bar H₂ at 80 °C to Z-2H (cis-β-methyl styrene), E-2H (trans-β-methyl styrene), 4H (n-propyl benzene) products. **c** Catalyst stability test tracking conversion (left) and selectivity (right) for 3, 10 h cycles using 0.5 mol% Ni₃Zn NCs, 1 bar H₂, 80 °C.

(Fig. 4b, Supplementary Fig. 24), showing constant product distributions (Supplementary Table 9). The spent NCs (Supplementary Fig. 25) remain monodisperse and retain their size, indicating that Ni₃Zn NCs are durable as catalysts. Concretely, the recycle tests were performed as follows: right after the hydrogenation experiment, the autoclave was opened, an aliquot was taken for product analysis, and 1 mL 1-phenyl-1-propyne stock solution (1 M, 1 mmol) was added to the reaction mixture. The reaction mixture was subjected to a subsequent hydrogenation run, (again 80 °C, 1 bar H₂, 0.5 mol% catalyst loading), and this procedure was repeated two times.

Taken together, non-precious bimetallic Ni₃Zn NCs are highly active for alkyne semihydrogenation catalysis, converting 1-phenyl-1-propyne selectively at hydrogen pressures as low as one bar (allowing for instance semihydrogenation reactions at small scale in laboratories with H₂-balloons). Such low hydrogenation pressures are usually only observed for noble metal catalysts (Supplementary Table 10). A disadvantage of the Ni₃Zn catalyst is its air-sensitivity, such that care must be taken to not oxidize the catalyst, which is however possible in academic and industrial settings. Despite their air-sensitivity, non-precious metal catalysts, such as Ni-Zn alloys, are attractive to pursue from an environmental impact and cost perspective, Supplementary Fig. 26. Many industrial players in the field of catalysis are looking for opportunities to decrease the CO₂-footprint and environmental impact of their processes and innovative catalyst designs based on non-precious metals offer an attractive solution regarding this.

### Impact of amine-termination of nanocrystal surfaces

The reactivity and durability of the Ni₃Zn NCs indicate that amine surface chemistry is non-poisoning and likely robust. To gain insight into the behavior and role of amine ligands on the Ni₃Zn NCs, we use DFT-calculations. We calculate the adsorption energies of 1-phenyl-1-propyne and 1-phenyl-1-propene with an ethylamine ligand co-adsorbed to the slab (Supplementary Fig. 27). The values are different in the presence and absence of an ethylamine ligand (Supplementary Table 11). As both the $E_{ads}$ of 1-phenyl-1-propene decreases and the difference in $E_{ads}$ between 1-phenyl-1-propyne or 1-phenyl-1-propene increases, we conclude that amine ligands will improve the selectivity of both Ni and Ni-alloy catalysts. We attribute the changes

in $E_{ads}$ to the carbon chain of an amine ligand sterically preventing the non-selective binding of the aromatic rings of 1-phenyl-1-propene and 1-phenyl-1-propyne to the slabs[43] (Supplementary Fig. 27).

### Substrate scope

To assess the overall catalytic performance of Ni₃Zn NCs for the semihydrogenation of alkynes, we next explore their use for various alkyne substrates (Fig. 5, Supplementary Table 12, and Supplementary Figs. 28–31). We primarily use hydrogenation conditions A (80 °C, 1 bar H₂, 10 h, 0.5 mol% catalyst), unless low rates of hydrogen-uptake are observed, in which case hydrogenation conditions B (80 °C, 5 bar H₂, 16 h, 0.5 mol% catalyst) are used.

Besides 1-phenyl-1-propyne, diphenylacetylene and phenylacetylene are also commonly studied in alkyne semihydrogenation catalysis. Compared to our model substrate, 1-phenyl-1-propyne, diphenylacetylene requires conditions B to convert to cis-stilbene, though with excellent selectivity (92%). Phenylacetylene is converted under conditions A to ethylbenzene with 94% yield, indicating that even milder conditions should be employed to slow down the reaction and obtain styrene selectively. As shown in Fig. 5b, the different steric properties of these substrates, as quantified by the buried volume of the alkyne, affect their reactivity for alkyne semihydrogenation.

To assess the impact of the electronic properties on hydrogenation activity independent from steric effects, we test a range of functionalized phenylacetylene-type substrates with comparable steric characteristics, Fig. 5c. We use Hammett substituent constants for para-substituted aryl molecules ($\sigma_p$) as a descriptor for the influence of electron-donating and electron-withdrawing substituents on catalytic reactivity[67]. We find that electron-withdrawing substituents such as chloro-, methylester-, cyano-, and nitro- substituents reduce reactivity, while electron-donating substituents like dimethylamino-, methoxy-, and methyl- increase reactivity. This trend indicates that an increase in electron density on the alkyne-function will increase the overlap between the substrate and the d-orbitals of the Ni-atoms, improving the binding interaction. The slightly lower reactivity observed for the substrates featuring dimethylamino- and methoxy-substituents compared to a methyl-substituent may be explained by the dimethylamino- and methoxy-substituents acting as ligands which bind competitively to the catalyst.

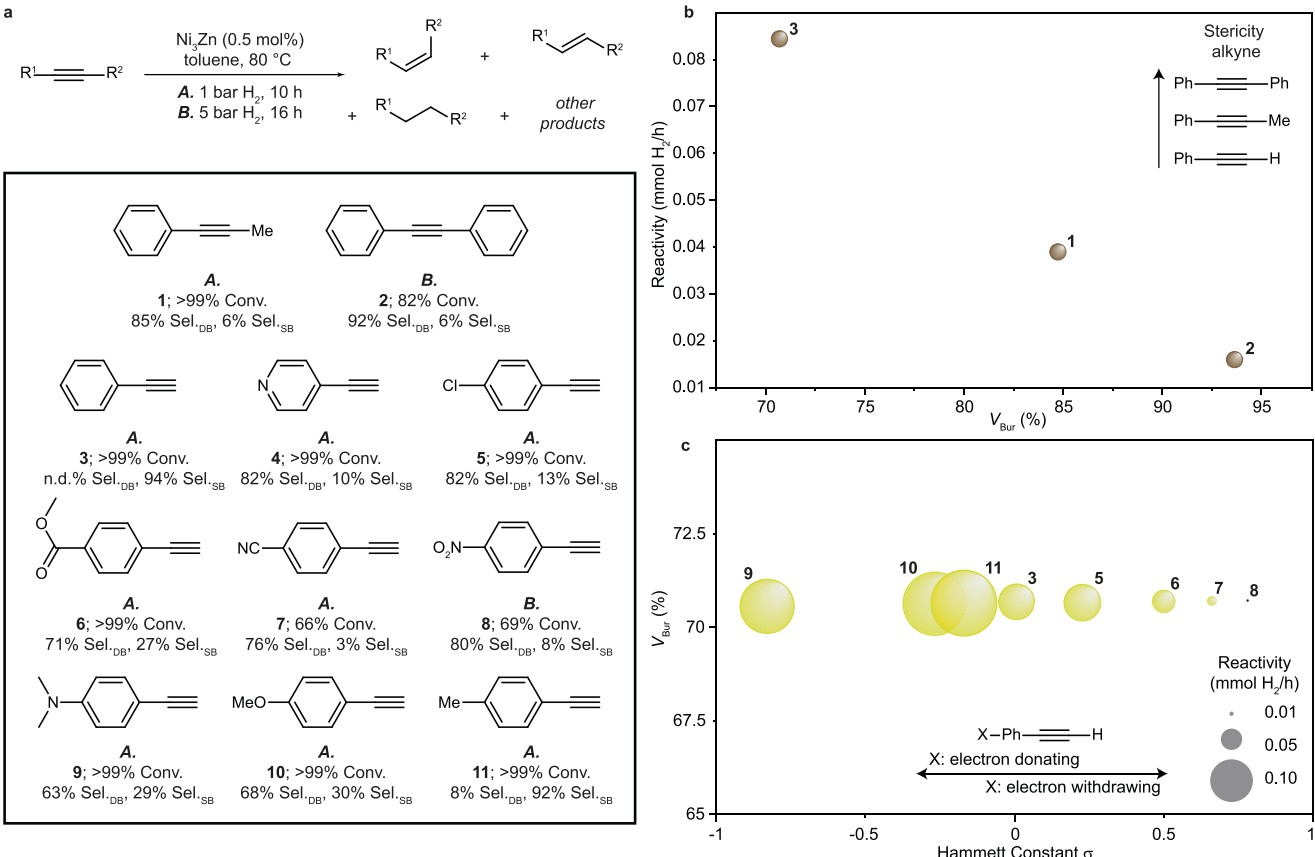

**Fig. 5 | Influence of alkyne steric and electronic properties. a** Semihydrogenation of different alkyne substrates catalyzed by Ni$_3$Zn NCs. Conv. conversion, Sel.$_{DB}$ selectivity for the *cis*-alkene (resulting from internal alkynes) or Sel.$_{DB}$ selectivity for the alkene (resulting from external alkynes), Sel.$_{SB}$ selectivity for the alkane.

Reactivity of the alkyne substrates as quantified by hydrogen uptake rate as a function of their **b** buried volumes $V_{Bur}$ (steric descriptor) and **c** Hammett substituent constants $\sigma_p$ (electronic descriptor) for para-substituted aryl molecules.

Expanding the substrate scope study (Fig. 6, Supplementary Table 13, and Supplementary Figs. 32 and 38) beyond phenylacetylene-type substrates functionalized with various substituents reveals that Ni$_3$Zn NCs show excellent functional group tolerance, chemoselectively converting alkynes bearing nitrile-, aryl-. methoxy-, ester-, halide-, amine-, pyridine-, nitro-, phthalimide-, and ketone-moieties. We attribute this to the beneficial alloying of Ni with Zn, as Zn was found to interact only weakly with functional groups on the substrates compared to Ni (hindering unselective co-adsorption of the functional groups to the catalysts), and the presence of oleylamine ligands on the nanocrystals, which provide a steric environment that also assists in preventing the unselective co-adsorption of functional groups on the substrates to the catalyst.

## Discussion

We developed a facile synthesis method for small and monodisperse Ni and Ni-X (X= Zn, Ga, In) NC catalysts, which we test here for alkyne semihydrogenation. We show that the oleylamine capping ligands are non-poisoning for alkyne semihydrogenation catalysis and are also likely to improve the selectivity of the reaction.

Comparing different catalysts compositions, we find that Ni$_3$Zn NCs are more active than Ni$_3$Ga and Ni$_3$In NCs, operating under mild conditions and at low loadings. We explain this with DFT-calculations, which reveal that Zn-atoms in the alloy contribute to the binding of alkyne substrates in contrast to Ga- or In-atoms. The Ni$_3$Zn NCs also displayed durability over multiple catalytic runs as well as good functional group tolerance, efficiently catalyzing the semihydrogenation of a wide range of alkyne substrates. The activity of the

catalyst increases with increasing electron density on the alkyne functionality.

The findings and synthesis method presented here can be used to further engineer bimetallic and ternary alloy NCs with improved catalytic properties. In particular, our findings highlight that active and selective semihydrogenation catalysts can be engineered through alloying metals that interact strongly with alkynes with metals that weakly adsorb alkynes but which do not participate in alkene bonding, as exemplified here for Ni-Zn. We hypothesize that this could explain the generally good performance of bimetallic catalysts consisting of active group 10 metals alloyed with elements from group 11 and 12 in the literature (e.g., PtCd, PtZn, PdAg, NiAu, etc.[30,42,49]).

## Methods

### Materials for nanocrystal synthesis

Bis(μ-dimethylamino)tetrakis(dimethylamino)digallium (Ga$_2$(NMe$_2$)$_6$, 98%), indium(III) chloride (InCl$_3$), zinc(II) chloride (ZnCl$_2$) and nickel(II) acetylacetonate (Ni(acac)$_2$, min. 95%) were purchased from STREM. Zinc bis[bis(trimethylsilyl)amide] (Zn[N(SiMe$_3$)$_2$]$_2$, 97%), lithium bis(-trimethylsilyl)amide (Li[N(SiMe$_3$)$_2$], 97%), tert-butylamine borane (tBAB, 97%), boron nitride (BN, 99.5%), methyl acetate (anhydrous, 99.5%), hexane (anhydrous, 95%), diethyl ether (Et$_2$O, anhydrous, ≥99.0%) pentane (anhydrous, ≥99%) and toluene (anhydrous, 99.8%) were purchased from Sigma Aldrich. Oleylamine (OAm, approximate C18 content 80–90%) and squalane (99%) were purchased from Fischer Scientific. Squalane and OAm were degassed under vacuum at a temperature of 110 °C for 120 min, cooled to room temperature and transferred to a N$_2$-filled glovebox. Boron nitride powder was dried at 250 °C

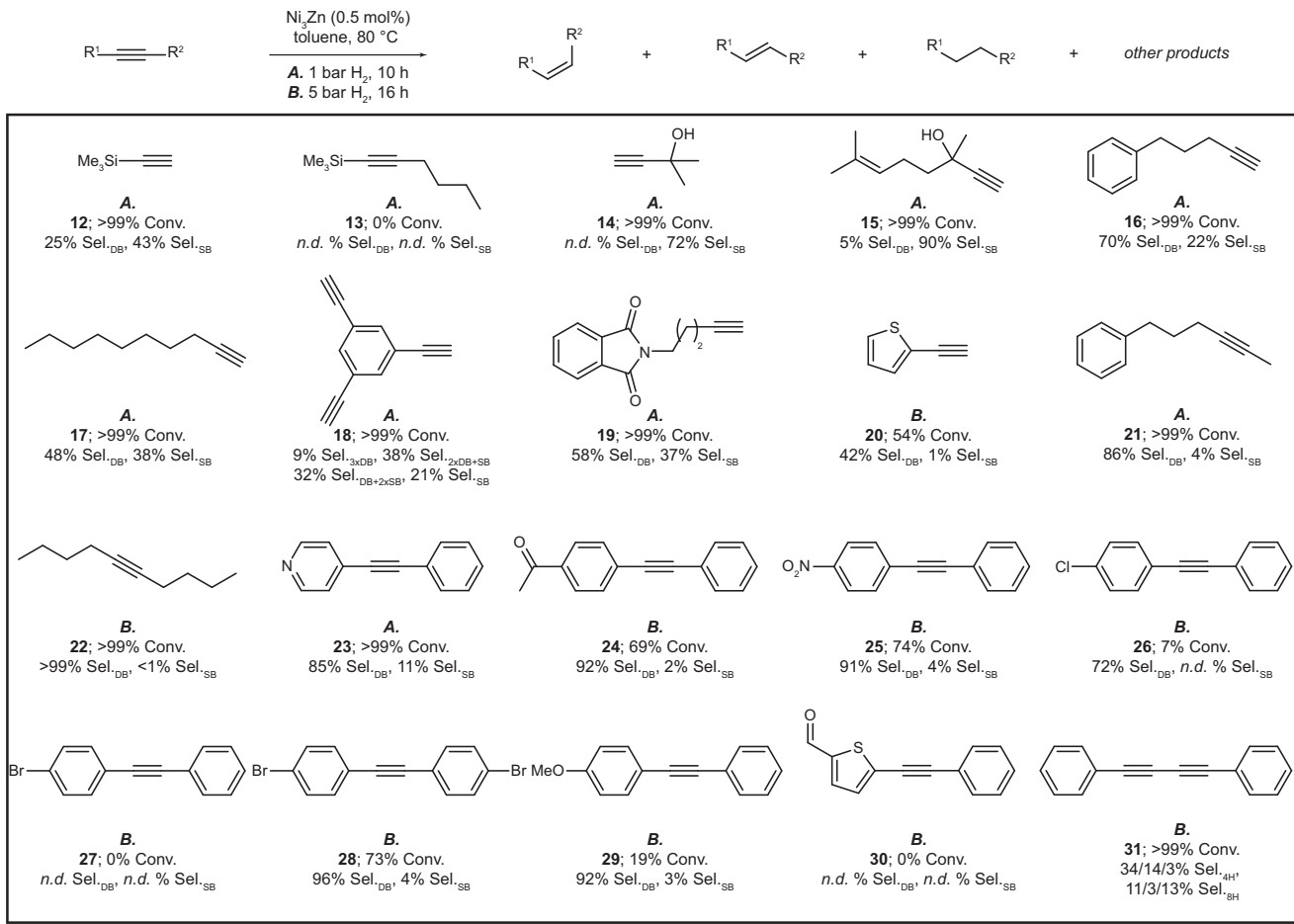

**Fig. 6 | Substrate scope experiments.** Second part of the substrate scope for the semihydrogenation of alkynes by Ni₃Zn NCs; Conv. conversion, Sel.DB selectivity for the *cis*-alkene (resulting from internal alkynes) or Sel.DB selectivity for the alkene (resulting from external alkynes), Sel.SB selectivity for the alkane.

under vacuum. Solvents were stored over molecular sieves (3 Å Sigma Aldrich) inside a N₂-filled glovebox. In[N(SiMe₃)₂]₃ was prepared according to Bürger et al. [68]. Similar to In[N(SiMe₃)₂]₃, Zn[N(SiMe₃)₂]₂ can also be prepared via this facile salt metathesis reaction[68]. We found that purification-by-distillation of both commercially received and synthesized Zn[N(SiMe₃)₂]₂ improved the reproducibility of the catalytic performance of the as-synthesized Ni-Zn NCs. All other chemicals were used as received. All experiments were carried-out under N₂-atmosphere using conventional Schlenk-line techniques or in a N₂-filled glovebox. Nanocrystals were handled and stored inside a N₂-filled glovebox.

## Preparation of In[N(SiMe₃)₂]₃

A solution of Li[N(SiMe₃)₂] (15 mmol, 2509.9 mg) in 50 mL Et₂O was added dropwise to a stirred solution of InCl₃ (5 mmol, 1105.9 mg) in 150 mL Et₂O at room temperature and was then left stirring overnight (12 h) at room temperature. The solution was subsequently passed through a PTFE filter (0.2 μm) and then evacuated, resulting in a yellow-white colored solid. In[N(SiMe₃)₂]₃ was recrystallized using pentane, yielding a white powder (with a faint yellow hue).

## Synthesis of Ni Nanocrystals

3.2 ± 0.4 nm Ni seeds were synthesized employing a hot-injection procedure with tBAB as a reducing agent. Ni(acac)₂ (0.5 mmol, 128.5 mg) was loaded in a three-neck flask and dissolved in oleylamine (8 mL) at 120 °C at the Schlenk-line. The reaction mixture was then brought to 180 °C followed by the swift injection of a solution of tBAB (5 mmol, 434.9 mg) in OAm (4 mL), turning the reaction mixture

instantly into a black colored dispersion of Ni seeds. The reaction mixture was kept stirring for 10 min at a temperature of 180 °C. The reaction solution of Ni NCs was then cooled down to room temperature and either brought inside the glovebox for post-synthetic purification or evacuated on the Schlenk-line for 30 min prior to alloying the Ni NCs via amalgamation seeded growth.

## Synthesis of Ni-X (X = Zn, Ga, In) Bimetallic Nanocrystals

To synthesize bimetallic Ni-X NCs, a dispersion of Ni seeds, prepared as described above, was brought to a temperature 250 °C under N₂-flow. Next, a precursor solution of either (0.063 mmol, 24.3 mg), (0.167 mmol, 64.5 mg) or (0.5 mmol, 193.1 mg) Zn[N(SiMe₃)₂]₂, (0.0835 mmol, 33.7 mg) Ga₂(NMe₂)₆ or (0.167 mmol, 99.5 mg) In[N(SiMe₃)₂]₃ in 2 mL of squalane was injected in the Ni-seed solution for the synthesis of Ni₈Zn, Ni₃Zn, NiZn, Ni₃Ga and Ni₃In bimetallic nanocrystals respectively. The reaction mixture was kept at a temperature of 280 °C for 10 min and subsequently cooled down to room temperature and brought inside the glovebox for purification. To purify the nanocrystals, synthesis solutions were added to centrifuge tubes along with 80 mL of anhydrous methyl acetate. The nanocrystals were precipitated via centrifugation at 9000 rpm (12,298 g) for 5 min. The supernatant was discarded, and the precipitated nanocrystals were redispersed in 10 mL of anhydrous toluene.

## Electron microscopy measurements

Transmission electron microscopy (TEM) images and Selected Area Electron Diffraction (SAED) images were captured with a Hitachi HT7700 microscope operating at 100 kV. High-angle annular dark-field

scanning TEM images (HAADF STEM), energy dispersive X-ray spectroscopy (STEM-EDX) elemental mappings and high-resolution TEM images (HRTEM) were collected with a FEI Talos microscope operating at 200 kV. Aberration corrected STEM images were taken with a Hitachi HD2700 microscope operating at 200 kV. In order to prepare the nanocrystal samples for TEM measurements, nanocrystals were further purified with another precipitation-centrifugation cycle using anhydrous methyl acetate. Diluted colloidal dispersions of nanocrystals in anhydrous hexane were drop-cast onto a carbon grid (TedPella, ultrathin carbon film on a lacey carbon support). TEM grids with nanocrystals were transported inertly to the electron microscopes, limiting the amount of air-exposure to ca. 1 min upon mounting the grids onto the electron microscopy holders. Nanoparticle sizes were measured by counting the diameters of 300 nanoparticles in the acquired TEM images using ImageJ software. Size deviations were obtained from formula (1), employing a Gaussian fit of the obtained size histograms:

$$s = \frac{fwhm}{2} \qquad (1)$$

where fwhm is the full-width at half-maximum of the Gaussian fit.

## Powder X-ray diffraction (XRD) measurements

Powder XRD patterns were measured on a Rigaku SmartLab 9 kW System, equipped with a rotating Cu anode and a 2D solid-state detector HyPix-3000 SL. Colloidal dispersions of the as-synthesized nanocrystals were loaded in quartz capillaries (1 mm diameter) inside a glovebox, sealed with epoxy resin, and measured in parallel beam geometry.

## Elemental analysis

To determine the elemental composition of the nanocrystals, Scanning Electron Microscopy Energy-Dispersive X-ray (SEM-EDX) Spectroscopy measurements were carried out with a FEI Quanta 200 FEG SEM microscope, operating at 30 keV. Nanocrystal synthesis yields of 100% were confirmed by measuring the Ni-content of the dispersions with Inductively Coupled Plasma Optical Emission Spectroscopy (ICP-OES). To perform ICP-OES measurements, diluted solutions of the nanocrystals (1–10 ppm Ni) were digested in aqua regia and measured three times on an Agilent 720S system. Tubes of the ICP-OES device were flushed for 1 min with diluted aqua regia between sample measurements.

## X-ray photoelectron spectroscopy (XPS) measurements

XPS analysis was performed using the SIGMA II instrument equipped with an Al/Mg twin anode. The vacuum in the chamber was between $10^{-8}$ and $10^{-7}$ mbar during measurements. The measurements were carried out with an Al-Kα source (energy = 1486.6 eV, 0.1 eV step size) in LAXPS mode with a pass energy of 16 eV. The curve-fitting was performed with Shirley background and lineshape GL(30). C1s was used for referencing at 284.8 eV. Peakfitting was performed in CasaXPS software. Relative Sensitivity Factors (RSFs) build-in in CasaXPS were used for the semi-quantitative analysis of surface species. All Ni₃X NCs show approximately 50% surface content of the metal X instead of the 25% expected. This is consistent with Scanning-Tunnelling Electron Microscopy (STEM) Energy Dispersive X-Ray (EDX) elemental mapping, Supplementary Fig. 17, which indicates a Zn-enriched surface on Ni₃Zn NCs. The fact that some Zn, Ga, and In is drawn to the surface of the alloy NCs might be explained by the post-synthetic purification procedure with methyl acetate and the capping of the surface with oleylamine ligands. The small differences in surface content of Zn, Ga, and In agree with the solubility of the elements in face-centered cubic (fcc) Ni, Supplementary Fig. 18. We therefore use model slabs of Ni and Ni-alloys with a stoichiometry of the upper two layers set to a 50:50 ratio (and random mixing) in the calculations. In order to prepare the

nanocrystal samples for XPS-measurements, nanocrystals were further purified employing an additional precipitation-centrifugation cycle using anhydrous MeOAc. Nanocrystals were then redispersed in anhydrous hexane and drop-cast onto Au-coated Si-wafers (50 nm Au). To further remove ligands, the wafers were submersed three times in anhydrous MeOAc. Ga₂O₃ and In₂O₃ references were obtained by oxidizing Ni-Ga and Ni-In NCs respectively in air.

## Density functional theory (DFT) calculations

Considering the DFT calculations for the adsorption of molecules on the modeled slabs, the adsorption energy ($\Delta E_{ads}$) is calculated as (2):

$$\Delta E_{Ads} = E_{Adsorbate\ on\ slab\ surface} - E_{Slab} - E_{Adsorbate} \qquad (2)$$

where $E_{Slab}$ and $E_{Adsorbate}$ are the total energy of the fully relaxed slab and adsorbates in their ground states, and $E_{Adsorbate\ on\ slab\ surface}$ is the total energy of the relaxed adsorbates-on-slab-surface system, where the bottom two layers of the slab were fixed at the relaxed position in the absence of adsorbates, and the top four layers and the adsorbates were fully relaxed. Further information considering the performed DFT calculations is provided in the Supplementary Information.

## Steric descriptors of alkynes

We use buried volume ($V_{bur}$) as a parameter to describe the steric properties of the alkynes, see Supplementary Information (Supplementary Fig. 39).

## Materials for catalytic tests

Unless otherwise noted, all synthetic manipulations were performed under inert (Ar) atmosphere, using dry, oxygen-free solvents, either inside a glovebox, or using appropriate Schlenk techniques, with oven-dried glassware equipped with Teflon-coated magnetic stirring bars. Lindlar catalyst, Palladium, 5% on CaCO₃, lead-poisoned, was purchased from STREM. Toluene (Sigma Aldrich, 99.9%) was purified by passage through double solvent purification alumina columns (Mbraun) and stored over activated Selexsorb CD® (BASF). Unless otherwise stated all commercially available chemicals were used without further purification. n-Tridecane (TCI, >99%) was stirred over sodium for one day, vacuum transferred and taken into an Argon-filled glovebox. Considering the alkyne substrates, 1-phenyl-1-propyne (99%), 4-ethynylbenzonitrile (97%), 1-ethynyl-4-dimethylaniline (97%), 1-decyne (98%), 2-methyl-3-butyn-2-ol (98%), hex-1-yn-1-yltrimethylsilane (99%), N-(4-pentynyl)phthalimide (97%), 4-ethynylpyridine (95 %), 1-chloro-4-(phenylethynyl)-benzene (98 %) and 1-ethynyl-4-nitro-benzene (97%) were purchased from Sigma-Aldrich. Phenylacetylene (98 %), 4-methyl-phenyl-acetylene (97%) and diphenylacetylene (99%) were purchased from Acros Organics. 4-Chloro-phenylacetylene (98%) and methyl-4-ethynylbenzoate (98%) were purchased from Fluorochem. 4-Methoxyphenylacetylene, (trimethylsilyl)-acetylene and 2-ethynylthiophene (95+ %) were purchased from Apollo. 5-Phenyl-1-pentyne (98%), 6-phenyl-2-hexyne (99%), 4-(phenylethynyl)-acetophenone (97%), 1-bromo-4-(phenylethynyl)-benzene (98 %) and 1,4-diphenylbutadiyne (99 %) were purchased from ABCR. 3,7-Dimethyl-6-octen-1-yn-3-ol (>98%), 1,3,5-triethynylbenzene (>98%) and 1,2-bis(4-bromophenyl) ethyne (>98 %) were purchased from TCI. 1-Methoxy-4-(phenylethynyl)-benzene was purchased from Rieke Metals. 4-(2-Phenyleth-1-ynyl)-thiophene-2-carbaldehyde (97%) was purchased from Maybridge (UK). 5-decyne (98%) was purchased from Alfa Aesar. 4-(Phenylethynyl)-pyridine and 1-nitro-4-(phenylethynyl)-benzene were synthesized using an adapted literature procedure for the copper-free Sonogashira cross-coupling reaction[69]. All liquid alkynes were degassed using 3 freeze-pump-thaw-cycles and filtered through a pad of activated alumina (high vacuum at 300 °C for >12 h). The filtration was carried out in a glovebox and repeated until the pad of alumina remained colorless. 1-Phenyl-1-propyne was stirred over

CaH$_2$ for one day and vacuum transferred, before filtration over activated alumina. All solid alkynes were used as received.

## Gas chromatography

GC measurements were performed on a Shimadzu-QP 2010 Ultra using an HP-5 column (30 m × 250 μm × 0.25 μm), with a column flow of 4.7 mL/min using helium as carrier gas, a FID (40:400 H$_2$/Air, 30 mL/min He makeup flow) and a MS operating with scan rate of 0.1 s from 50 to 350 m/z. Sample injection was done by a Shimadzu AOC-5000 Plus autosampler, 1 μL in split mode (split ratio 30:1) with an injection port temperature of 310 °C. The longest method used for the analysis of the heaviest substrates consisted of the following temperature program: 4 min at 35 °C, then a ramp of 5 °C/min to 45 °C, hold for 3 min, ramp of 25 °C/min to 95 °C, hold for 1 min, ramp of 5 °C/min to 275 °C and hold for 12 min (total 60 min). Aliquots (0.1 mL) of reaction solutions were diluted with toluene (0.4 mL).

GC quantification was performed with the FID chromatogram integrated peak areas calibrated against each MS-detected compound's response factor relative to n-tridecane internal standard, under the assumption that response factors for the respective alkynes, alkenes and alkanes are the same. The quantification gave amounts of formed products, as well as remaining substrate that were used to calculate reported values of conversion (X = conversion) and selectivity (S = selectivity).

$$X_{Substrate} = \frac{n_{Substrate}^{Initial} - n_{Substrate}}{n_{Substrate}^{Initial}} *100\% \quad (3)$$

$$S_{Semihydrogenation} = \frac{n_{All\ alkenes}}{n_{Total\ products}} \quad (4)$$

$$S_{Alkene} = \frac{n_{cis-Alkenes}}{n_{Total\ products}} \quad (5)$$

$$Carbon\ Balance = \frac{n_{Substrate}^{Initial}}{n_{Total\ products} + n_{Substrate}} \quad (6)$$

The amount of total products formed is the sum of MS-detected products, including cis/trans-alkenes, alkene chain isomers dimers/trimers and alkanes. In general, the carbon balance closes within a few percent (2-3%) of 100%. Carbon balances below 97% are attributed to formation of oligomers and/or strong binding of products or substrates to the catalyst that are not detectable via GC. The reported selectivity values are lower estimates, calculated under the assumption that oligomers are responsible for the <100% carbon balances and are hence considered as products.

## High pressure reactor setup for catalytic tests

The catalytic hydrogenation experiments were performed at the High Throughput Experimentation facility of the ETH Zürich (HTE@ETH) on an Endeavor autoclave (Biotage) operated inside a glovebox (Supplementary Fig. 8). The H$_2$-feed was passed through activated Cu/Al$_2$O$_3$ sorbent (BASF) and activated 4 Å molecular sieves before entering the autoclave system. Pressures are given in bar (g).

## Alkyne hydrogenation experiments

Inside a glovebox, a glass liner was loaded with 1 mL alkyne stock solution (1 M, 1 mmol) in toluene containing n-tridecane as internal standard (0.1 M, 0.1 mmol). Subsequently, a stock solution of the colloidal nanocrystals in toluene was added and the mixture was further diluted to 3 mL of toluene (resulting in 0.5 mol% catalyst loading). Up to 8 liners were loaded into an 8-parallel autoclave (Endeavor, Biotage) operated inside the glovebox. The reactors were leak checked with

5.4 bar argon, depressurized, and subsequently heated and pressurized with hydrogen to the desired conditions. The stirring rate was set to 500 rpm. To stop the reaction, the gas supply was stopped, the reactors were closed and were cooled down to 50 °C or for a maximum of 90 min, before venting and purging 3 times with 5.4 bar argon. Quantification of the reaction mixtures by GC analysis provided conversion and selectivity values. Two hydrogenation procedures were in general employed:

- Hydrogenation Procedure A: 0.5 mol% catalyst, 1 bar H$_2$, 80 °C, 10 h.
- Hydrogenation Procedure B: 0.5 mol% catalyst, 5 bar H$_2$, 80 °C, 16 h.

All gas uptake curves were recorded using the Endeavor parallel batch reactor system from Biotage with a data polling interval of 1 s. Although pressurization and recording start immediately, each curve is plotted starting from the time of reaching the desired reaction temperature. All gas uptakes are normalized to the expected H$_2$ consumption as determined by GC-FID analysis: 1 mol per semihydrogenation products and oligomerization products, and 2 mmol per overhydrogenation products (in case of 0% conversion, the H$_2$-uptake plots show a horizontal line, as no normalization is possible). The molarity of the reaction is indicated by the horizontal gray dashed line.

The uptake curves were smoothened by applying a Savitzky-Golay filter with a window length of 2501 and a 2$^{nd}$ order polynomial. In order to determine the hydrogenation rate, the inflection point(s) of each curve is/are determined by calculating the second derivative of the smoothed function. The hydrogenation rate is then given by the slope of the line going through the first timepoint and the inflection point, the inflection point and the last timepoint (typically in low conversion cases), or through two inflection points.

## Data availability

Source data are provided with this paper and are uploaded to the ETH Zurich Research Collection (https://doi.org/10.3929/ethz-b-000727149). Source data are provided with this paper.

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

## Acknowledgements
Electron microscopy measurements were performed at the Scientific Center for Optical and Electron Microscopy (ScopeM) of the Swiss Federal Institute of Technology. DFT simulations were performed on the ETH Zürich Euler cluster. The authors acknowledge for discussions and assistance with respectively XAS-measurements: Jan Alfke, Olga Safonova and Simon Wintersteller; SEM EDX: Annelies Landuyt and Olesya Yarema; LCA: Leopold Peiseler; Graphics: Maximilian Jansen. Manuscript revisions: Annelies Landuyt. The authors also want to thank Matthew S. Sigman, Shahar Dery and Christian Ehinger for fruitful discussions. Relevant funding: ETH Zürich, Research Grant ETH-37 18-2: *J.C.*; National Research Fund Luxembourg, AFR Individual Ph.D. Grant 12516655: *J.D.J.S.*; European Research Council, grant agreement no. 852751: *M.Y.*

## Author contributions
*J.C.* and *J.D.J.S.* contributed equally to this work; *J.C.* performed the synthesis and characterization of the nanocrystals; *S.B.X.Y.Z.* performed the XPS measurements in collaboration with *J.C.*; *Y.X.* performed DFT calculations in collaboration with *J.C.* under the supervision of *N.Y.*; Initial conceptualization: *J.C.*, *J.D.J.S.*, and *S.R.D.*; overall supervision: *C.C.* and *V.W.*; Further support: *M.Y.*; *J.C.* and *J.D.J.S.* designed and performed the substrate scope; Initial drafts were written by *J.C.* and *J.D.J.S.*; Further editing of the manuscript was conducted by all authors; All authors have given approval to the final version of the manuscript.

## Funding

## Competing interests
The authors declare no competing interests.
