## [Peer Review File · Nature Communications]

Earth-Abundant Ni-Zn Nanocrystals for Efficient Alkyne Semihydrogenation Catalysis

Corresponding Author: Professor Vanessa Wood

Version 0:

Reviewer comments:

Reviewer #1

(Remarks to the Author)

The article "Earth-Abundant Ni-Zn Nanocrystals for Efficient Alkyne Semihydrogenation Catalysis" provides a detailed examination of the development and use of Ni-Zn nanocrystals as catalysts for alkyne semihydrogenation. The research introduces an innovative catalyst composed of earth-abundant materials, addressing both environmental and economic concerns associated with scarcer and costlier metals. By utilizing less toxic and more readily available materials, the study supports sustainable practices in chemical manufacturing.

However, it is not appropriate to use technical terms like "high-throughput experimentation" in the abstract. While the paper effectively discusses the synthesis and performance of the catalysts, it lacks detailed information on the economic aspects and scalability of the production process, which are vital for industrial application. Additionally, although it references current catalysts like the Lindlar catalyst, a more comprehensive comparison regarding cost, efficiency, and environmental impact would better substantiate the advantages of the Ni-Zn nanocrystals. The article predominantly highlights the positive attributes of the Ni-Zn nanocrystals. A more thorough discussion of any limitations or challenges encountered during the experiments would offer a more balanced perspective. If feasible, incorporating coverage effects into the DFT calculations would render the results more accurate and enhance the overall quality of the paper.

Reviewer #2

(Remarks to the Author)

The authors reported the successful synthesis of small Ni and bimetallic Ni-X (X= Zn, Ga, In) NCs for alkyne semihydrogenation catalysis. The Ni₃Zn NCs are reactive and selective under mild reaction conditions and at low loadings. And use theoretical calculations to demonstrate the reaction mechanism. However, the structures of the nanocrystals are unclear although the author provided XANES, TEM, XRD, and XPS results. More characterization and discussions should be provided. This manuscript has some interesting results, but before it can seriously be considered for publication, major revisions are required. Specific comments for the authors to address are provided below.

1. The XRD curves do not match with the particle size, as the Ni₃Ga nanocrystals present higher intensity in XRD curve compared with other samples. However, the particle size of Ni₃Ga (3.6 nm) < Ni₃In (3.9 nm) < NiZn (4.0 nm).
2. More EDS-mappings should be provided for the bimetallic Ni-based alloys, the composition of a single nanocrystal is not clear in Fig S16.
3. In Fig. S6, the binding energy of Ni 2p for monometallic Ni, Ni₈Zn, Ni₃Zn, NiZn, Ni₃Ga and Ni₃In change from 853.6 to 850.7 eV. The 853.6 eV should be ascribed to Ni²⁺ not metal Ni, demonstrating that the Ni crystals might be coated with NiO_x layers. In addition, why the binding energy of Ni in Ni₈Zn lower than Ni₃Zn and NiZn with the increasing of Zn content as the author pointed out that charge transfer from Zn to Ni. The authors should reorganize this section.
4. Why do the authors use H₂ uptake not the yield to demonstrate the catalytic performance.
5. The cycle test in Fig. 4 is under the full conversion. The data does not provide clear information on the stability of the catalysts. If the conversion is 100%, the catalyst may deactivate without any visible change of in the product composition. How was the recycle tests conducted? The authors should provide details.
6. The lattice spacing in Fig S4 is not right.
7. The electronic and geometric structures of the nanocrystals should be added in the manuscript, no description for Fig. S1-5 was included in the manuscript.
8. Some PdPb alloys were also proved good candidates for the semihydrogenation reactions even with the addition of Pb.

Reviewer #3

(Remarks to the Author)

The manuscript of Coperet, Wood et al. presents an interesting study on the synthesis of Ni-Zn nanocrystals with various compositions (Ni₈Zn, Ni₃Zn, NiZn) and their catalytic performance in the semihydrogenation of alkynes. The findings suggest that Ni₃Zn is the most active phase with sufficient selectivity, outperforming Ni₃Ga and Ni₃In. While the scientific work is generally compelling, several critical issues regarding the conclusions drawn from the data need to be addressed for the manuscript to be suitable for publication.

Comparison with Key Literature (Ref 27):

The manuscript cites a key publication (Ref 27) that comprehensively studies the semihydrogenation of acetylene to ethylene using M-E intermetallic materials as catalysts, including Ni-Zn. According to Ref 27, NiZn₃ is computationally and experimentally the best catalyst with the optimal activity/selectivity tradeoff among all investigated materials.

The authors of the current manuscript conclude that Ni₃Zn is superior to NiZn, which differs from the findings of Ref 27. This discrepancy is not adequately discussed or explained in the manuscript. A detailed paragraph should be added, discussing this. Additionally, given the importance of Zn-rich phases as highlighted in Ref 27, it is crucial for the authors to extend their investigation to include the NiZn₃ phase. This will provide a more comprehensive understanding of the catalytic performance across different Zn compositions.

Potential Surface Oxidation:

Activity differences of catalyst materials with different Zn content might also be due to partial oxidation of Zn at the catalyst surface, either during nanoparticle preparation or catalysis. However, the manuscript lacks convincing evidence for the absence of oxygen in the as-synthesized nanocrystals. The authors should provide more thorough characterization to confirm the absence of surface oxidation. XPS for oxygen detection, XAS, or infrared spectroscopy (IR) could be employed. It would be also very important to thoroughly characterize the nanoparticles after catalysis to determine if their nature changes during the reaction.

BTW, the HAADF plots in Fig. 16 are insufficient to accurately determine the distribution of Ni and Zn. Separate plots for Ni and Zn are necessary to visualize their respective distributions clearly.

Comparison with State-of-the-Art Catalysts:

The manuscript should include an extended discussion comparing the performance of the Ni-Zn nanocrystals with state-of-the-art catalysts for both acetylene hydrogenation in ethylene feedstocks and Z-selective semihydrogenation of alkynes in fine chemical synthesis.

A detailed comparison regarding activity, selectivity, and tolerance towards functional groups will provide valuable insights into potential applications of the nanocrystal catalysts and their advantages over existing technologies.

Mechanistic Aspects and Recent Literature:

The manuscript includes calculated mechanistic aspects (e.g., SI, Fig. 22) that would benefit from being compared to recent literature confirming catalysis on Ni₃ triangular sites, such as the publication in Angew. Chem. Int. Ed. 2023, 62, e202308790.

In summary, while the reported science is fundamentally interesting, the manuscript requires significant revisions to address the issues mentioned above. These include a thorough discussion of contradictory findings with key literature (especially Ref. 27), extension of the investigation to more Zn-rich phases (e.g. NiZn₃), comprehensive characterization of the nanocrystals before and after catalysis and sufficient evidence excluding surface oxidation.

Recommendation:

Reconsideration after major revision.

Reviewer #4

(Remarks to the Author)

Disclaimer up front: This is a referee from the field of homogeneous catalysis, I therefore cannot comment on the preparation and characterization of the nanocrystals used for the catalytic experiments. In a similar vein, the reader, in the eyes of this referee, could be guided a bit better with respect to which types of catalysts generally perform well from the homogeneous as well as the heterogeneous field. Next to the precious metal catalysts, there have been plenty of 3d elements that can be employed as catalytically active center.

The present paper describes the use of Ni/Zn nanocrystals (which have been prepared by an amalgamation seeded growth technique) and their use in catalytic alkyne semihydrogenations. Generally, the quest in these transformations is to reach high selectivity for the alkene (and stereoselectivity) along with high catalytic performance at low H₂ pressure. Over the last decade, several steps ahead in terms of stereo- and chemoselectivity have been made especially with the use of 3d metals, but the pressure limitation is still challenging. In this light, the present work makes an important contribution, as the nanocrystals are catalytically active at 1 bar H₂ pressure. The authors claim that a variety of alkynes can be transformed to the corresponding alkenes. The high selectivity and reactivity is backed up by computational data.

Considering the substrate scope and the selectivities for the individual reactions: How are the stereo- and chemoselectivities determined? The authors claim that there has been GC analysis of all reactions, but there is no data supporting this in the SI.

While for the standard reactions with phenylacetylene, the corresponding Z and E alkenes are commercially available and can be used as standards, this is not the case for most of the substrates shown. One can also not deduce that one particular stereoisomer will always appear first in the chromatogram. Therefore, data is needed to support the claims made in the paper. Furthermore, many of the substrates carry functional groups that also may be reduced, such as ketones, aldehydes, nitro and ester groups. Can overreductions be excluded? How has this been determined? Have the authors observed any protodehalogenation with the bromides?

In Fig 5a and Fig 6 the authors speak of selectivity for the Z-alkene for most of the substrates. However, the large majority here are terminal alkynes which cannot deliver E or Z products. Therefore, the selectivity indication makes no sense. (Or have there been reactions with D₂ to determine the E / Z selectivity?)

Catalytic semihydrogenations of 1,3-diyne are rare (see the works by Kirchner and Teichert, for example), and the stereoselectivity would be of interest here. Which stereoisomers are formed with the nanocrystals?

As far as this referee can see, no substrate has been isolated and the yield has been determined. Can the authors exclude a typical cyclotrimerization of alkynes as side-reaction? This is a commonly observed process reducing the overall yield.

Overall, this referee is of the opinion that the new type of catalyst described is definitely of high interest for the community. Before publication, the data to back up the claims in the present manuscript should be given.

Version 1:

Reviewer comments:

Reviewer #1

(Remarks to the Author)
Accept.

Reviewer #2

(Remarks to the Author)
I have no questions, and I agree the publication in Nature Communications.

Reviewer #3

(Remarks to the Author)

I would like to begin by acknowledging the authors for addressing the majority of my comments in a satisfactory manner. However, I still have two comments, primarily regarding the explanations related to the surface composition of the nanocrystals.

First, I appreciate the authors' response to my initial concern about the catalytic performance of Ni₃Zn being seemingly in contradiction to the literature. They propose that the surface of the Ni₃Zn nanocrystals has a near NiZn stoichiometry, supported by XPS data. Indeed, there is strong alignment between both the Ni(2p_{3/2}) and Zn(2p_{3/2}) peaks for Ni₃Zn and NiZn NCs, which suggests a similar surface composition. The STEM/EDX images (SI Fig. 16) show an even distribution of Ni and Zn over a larger area, to my opinion they do not provide detailed insight into the elemental distribution within individual particles due to insufficient resolution (at least as depicted in SI Fig. 16). I would like to note that it appears to me there is an error in this figure, as the images for Ni₃Zn and NiZn display what seems to be the same object.

Second, since the surface compositions of Ni₃Zn and NiZn NCs are similar, it is noteworthy that their catalytic performance differs so significantly. A short discussion about that should be added to the main text.

In summary, I find the manuscript to be much improved after revision, and I appreciate the authors' efforts in addressing most of my concerns. The surface characterization of the Ni₃Zn NCs clarifies the apparent contradiction with the literature. I recommend accepting the manuscript for publication after these minor revisions.

Reviewer #4

(Remarks to the Author)

In the current version, the authors submit a significantly improved and re-worked version of the manuscript. Most of my comments to the previous version of the paper have been properly addressed. The key problem remains the claim of the selectivities (stereo- and chemoselectivity) of the alkyne semihydrogenation process: the authors describe at length that their analysis of yield and selectivity has been carried out by GC-MS analysis. While gas chromatography alone with some detectors is a quantitative technique when internal standards and calibration curves are used, GC-MS is not. An important reason for this is the non-linear behaviour of the molecules during ionization. Therefore, no precise analysis of the abovementioned parameters can be drawn from GC-MS.

As GCMS has been wrongly used as key method and the authors still do not show any chromatograms or spectra, the important claims of yields and selectivities - crucial descriptors for new and competitive catalysts - are not substantiated by proper data. Therefore, this referee suggests the rejection of the manuscript in its present form.

Version 2:

Reviewer comments:

Reviewer #1

(Remarks to the Author)

Most comments to the previous version of the paper have been properly addressed. I recommend accepting the paper.

Reviewer #2

(Remarks to the Author)

I agree this paper published in Nature Communications.

Reviewer #3

(Remarks to the Author)

I have reviewed the responses of the authors as well as the revisions made to the manuscript. I am satisfied with the changes and believe the authors have addressed all concerns raised. I recommend the manuscript for publication in its current form.

Reviewer #4

(Remarks to the Author)

The authors have now responded to the referees comments and have clarified that they have indeed used GC-FID-MS as a method to quantify their results and in this revision there is one single example in which they show how the data was analyzed and how the selectivities of the alkyne (semi)hydrogenation was obtained.

From my perspective, in order to assess the selectivity, purity and even identity of the 20+ substrates with quite some structural diversity and different functional groups, I would have expected characterization of the products. I would have expected at least the GC traces for the individual products, as there is no characterization of the products anyways.

I leave the decision to the editors as to whether the provided data is sufficient to back up the claims made in the paper and whether or not one should believe that the claims made in the paper are true without any data for the individual molecules prepared.

Response to Reviewers

Reviewer 1.

“The article "Earth-Abundant Ni-Zn Nanocrystals for Efficient Alkyne Semihydrogenation Catalysis" provides a detailed examination of the development and use of Ni-Zn nanocrystals as catalysts for alkyne semihydrogenation. The research introduces an innovative catalyst composed of earth-abundant materials, addressing both environmental and economic concerns associated with scarcer and costlier metals. By utilizing less toxic and more readily available materials, the study supports sustainable practices in chemical manufacturing. However, it is not appropriate to use technical terms like "high-throughput experimentation" in the abstract. While the paper effectively discusses the synthesis and performance of the catalysts, it lacks detailed information on the economic aspects and scalability of the production process, which are vital for industrial application. Additionally, although it references current catalysts like the Lindlar catalyst, a more comprehensive comparison regarding cost, efficiency, and environmental impact would better substantiate the advantages of the Ni-Zn nanocrystals. The article predominantly highlights the positive attributes of the Ni-Zn nanocrystals. A more thorough discussion of any limitations or challenges encountered during the experiments would offer a more balanced perspective. If feasible, incorporating coverage effects into the DFT calculations would render the results more accurate and enhance the overall quality of the paper.”

- With regards to “high-throughput experimentation”, we were referring to the point that we tested many catalysts and substrates using the same procedures, but indeed the term is not a key point of discussion within the manuscript and can be removed from the abstract as suggested.
- We now include a section where we discuss environmental aspects as well as industrial potential for the presented catalyst compositions in the manuscript. We perform a Life-Cycle Analysis (LCA) and show the obtained environmental impact characterization factors of metals frequently incorporated in semihydrogenation catalyst alloys in Supplementary Figure 25 of the manuscript.^{1,2} This includes global warming potential (GWP) in kg CO₂-equivalents, environmental quality (EQ) in species per year, human health (HH) in disability-adjusted life years and natural resources (NR) in 2013 US dollars. The impact characterization factors were obtained using the ecoinvent3.10 database, and the ReCiPe 2016 v1.03 life-cycle impact assessment (LCIA) for midpoint levels (GWP) and endpoints levels (EQ, HH and NR) according to the hierarchist perspective.³ Furthermore, the new section discussing the environmental aspects of the catalysts also discusses the efficiency the catalyst and discusses the used colloidal synthetic method and its prospects regarding economics and scalability.
- In the new section discussing the economic and environmental aspects of the presented non-precious bimetallic catalysts, we now also discuss the challenges of the shown catalysts, which we believe are mainly within the oxophilicity of the catalysts.

- Many species are present during the catalytic reaction, including reactants, products, side-products, capping ligands and solvents. Assessing the influence of all these would be very computationally expensive and goes beyond the scope of this work. We believe that the DFT-calculations presented in this work capture most of the essential insights to be gained considering these systems. We also note that while many literature examples restrict themselves to simplified molecules, e.g. acetylene and ethylene, we also computed larger alkyne and alkene molecules, which were used as substrates in the experiments, e.g., 1-phenyl-1-propyne. We show the influence of their chemical structure (e.g., aromatic rings that can non-selectively interact with the slabs) on catalysis and found that groups, such as aromatic rings, sterically interact with capping ligands on the nanocrystals (which results in capping ligands preventing the aromatic rings from co-adsorbing to the catalyst surface, ultimately resulting in higher selectivity of the catalyst).

Reviewer 2.

“The authors reported the successful synthesis of small Ni and bimetallic Ni-X (X= Zn, Ga, In) NCs for alkyne semihydrogenation catalysis. The Ni₃Zn NCs are reactive and selective under mild reaction conditions and at low loadings. And use theoretical calculations to demonstrate the reaction mechanism. However, the structures of the nanocrystals are unclear although the author provided XANES, TEM, XRD, and XPS results. More characterization and discussions should be provided. This manuscript has some interesting results, but before it can seriously be considered for publication, major revisions are required. Specific comments for the authors to address are provided below. 1. The XRD curves do not match with the particle size, as the Ni₃Ga nanocrystals present higher intensity in XRD curve compared with other samples. However, the particle size of Ni₃Ga (3.6 nm) < Ni₃In (3.9 nm) < NiZn (4.0 nm). 2. More EDS-mappings should be provided for the bimetallic Ni-based alloys, the composition of a single nanocrystal is not clear in Fig S16. 3. In Fig. S6, the binding energy of Ni 2p for monometallic Ni, Ni₈Zn, Ni₃Zn, NiZn, Ni₃Ga and Ni₃In change from 853.6 to 850.7 eV. The 853.6 eV should be ascribed to Ni²⁺ not metal Ni, demonstrating that the Ni crystals might be coated with NiO_x layers. In addition, why the binding energy of Ni in Ni₈Zn lower than Ni₃Zn and NiZn with the increasing of Zn content as the author pointed out that charge transfer from Zn to Ni. The authors should reorganize this section. 4. Why do the authors use H₂ uptake not the yield to demonstrate the catalytic performance. 5. The cycle test in Fig. 4 is under the full conversion. The data does not provide clear information on the stability of the catalysts. If the conversion is 100%, the catalyst may deactivate without any visible change of in the product composition. How was the recycle tests conducted? The authors should provide details. 6. The lattice spacing in Fig S4 is not right. 7. The electronic and geometric structures of the nanocrystals should be added in the manuscript, no description for Fig. S1-5 was included in the manuscript. 8. Some PdPb alloys were also proved good candidates for the semihydrogenation reactions even with the addition of Pb. *Angew. Chem. Int. Ed.*, 2015, 54, 8271; *Chem. Mater.*, 2018, 30, 6338; *Nano Res.*, 2022, 15, 4973.”

- 1. It is correct that the intensity of the reflections in the powder XRD patterns does not correlate with the trend in particle sizes. This is due to the polycrystallinity of the nanocrystals, which leads to characteristic broadening of the reflections in the powder XRD patterns.⁴ Ni-Ga NCs are slightly less polycrystalline compared to Ni-In and Ni-Zn and therefore show sharper reflections in the powder XRD patterns.
- 2. We now provide further Energy X-ray Dispersive Spectroscopy elemental mappings in Supplementary Fig. 16 as the reviewer suggests.
- 3. We provide below an XPS spectrum of air-exposed Ni NCs, which are thus partially oxidized on purpose. Compared to the XPS-spectra reported for the pristine as-synthesized NCs in the manuscript, these air exposed NCs feature 2 peaks, attributed to Ni(0) and Ni(II) species. The peak position of oxidized Ni is found at a binding energy of 855.4 eV (see figure and table below). The zerovalency of Ni for the non-air exposed NCs is also observable from the acquired XANES measurements shown in Supplementary Fig. 5 comparing Ni NCs to Ni and NiO references.

Ni 2p 3/2 [eV]	species
852.3	Ni(0)
855.4	Ni(II)
861.5	Ni satellite

In addition, the reviewer inquires why the binding energy peak position of Ni₈Zn is lower compared to Ni₃Zn and NiZn in the Ni 2p_{3/2} XPS spectra shown in Supplementary Fig. 6 (resp. 850.7 eV, 851.5 eV and 851.5 eV), which is not completely in line with charge transfer. We point out that the differences in binding energy peak positions are less as 1 eV and thus care must be taken to interpret these differences. Especially because

oleylamine ligands are only partially removed before XPS-measurements (so they do not contaminate the vacuum chamber). Oleylamine ligands also donate electrons to the metals (L-type ligands), which would lower the binding energy peak position in XPS-measurements (charge donation from oleylamine is also visible in the acquired XANES measurements shown in Supplementary Fig. 4, comparing oleylamine capped Ni NCs to a Ni-metal foil). It was probably the case that more oleylamine ligands were present on Ni₈Zn NCs compared to Ni₃Zn and NiZn NCs during the XPS-measurements (the noise in the Ni 2p_{3/2} XPS spectrum of Ni₈Zn is also a bit higher compared to Ni₃Zn and NiZn) and therefore we don't build a discussion in the manuscript on this.

- 4. The H₂-uptake is a useful parameter to describe the performance of the catalyst compared to yield. Yields were obtained post-reaction from GC-MS analysis of the reaction mixtures. We always stopped reactions after a fixed time (10 h in case of mild hydrogenation conditions A and 16 h in case of harsher hydrogenation conditions B). If yield would have been used as a performance parameter, a catalyst converting the reactant twice as fast (e.g., 4 h instead of 8 h) would appear to be equally active. The H₂-uptake was monitored over time and thus, the slope can be used as an activity parameter independent of when full conversion is taking place. Note that in addition, we normalize the H₂-uptake over the yield of formed products. This is to account for overhydrogenation (i.e. a catalyst that overhydrogenates, performing two hydrogenation steps during the reaction, would not appear to be twice as active as a selective catalyst hydrogenating the products at a similar rate). The derivation of H₂-uptake as a performance parameter is described in the methods section.
- 5. The catalyst recycling tests are performed by adding new alkynes just after full conversion was reached. There was no delay during which the catalyst could deactivate, though such deactivation is not to be expected. The procedure for the recycle tests was already described in the captions of Supplementary Fig. 23 and Supplementary Table 9, but we now include the description of the recycle tests in the main text as well.
- 6. We remeasured the d-spacing of the (111) lattice fringes visible in the high-resolution TEM image in Supplementary Fig. 4 and again found this to be 2.05 Å. We edited the markings on the HRTEM image of the d-spacing in Supplementary Fig. 4, as perhaps they were unclear.
- 7. We added a description for Supplementary Figs. 1-5 in the manuscript as suggested by the reviewer, discussing the geometric and electronic structure of the as-synthesized nanocrystals.
- 8. We thank the reviewer for directing us towards these references and have included them as references in the manuscript.

Reviewer 3.

“The manuscript of Coperet, Wood et al. presents an interesting study on the synthesis of Ni-Zn nanocrystals with various compositions (Ni₈Zn, Ni₃Zn, NiZn) and their catalytic performance in the semihydrogenation of alkynes. The findings suggest that Ni₃Zn is the most active phase with sufficient selectivity, outperforming Ni₃Ga and Ni₃In. While the scientific work is generally compelling, several critical issues regarding the conclusions drawn from the data need to be addressed for the manuscript to be suitable for publication.

Comparison with Key Literature (Ref 27): The manuscript cites a key publication (Ref 27) that comprehensively studies the semihydrogenation of acetylene to ethylene using M-E intermetallic materials as catalysts, including Ni-Zn. According to Ref 27, NiZn₃ is computationally and experimentally the best catalyst with the optimal activity/selectivity tradeoff among all investigated materials. The authors of the current manuscript conclude that Ni₃Zn is superior to NiZn, which differs from the findings of Ref 27. This discrepancy is not adequately discussed or explained in the manuscript. A detailed paragraph should be added, discussing this. Additionally, given the importance of Zn-rich phases as highlighted in Ref 27, it is crucial for the authors to extend their investigation to include the NiZn₃ phase. This will provide a more comprehensive understanding of the catalytic performance across different Zn compositions.

Potential Surface Oxidation: Activity differences of catalyst materials with different Zn content might also be due to partial oxidation of Zn at the catalyst surface, either during nanoparticle preparation or catalysis. However, the manuscript lacks convincing evidence for the absence of oxygen in the as-synthesized nanocrystals. The authors should provide more thorough characterization to confirm the absence of surface oxidation. XPS for oxygen detection, XAS, or infrared spectroscopy (IR) could be employed. It would be also very important to thoroughly characterize the nanoparticles after catalysis to determine if their nature changes during the reaction. BTW, the HAADF plots in Fig. 16 are insufficient to accurately determine the distribution of Ni and Zn. Separate plots for Ni and Zn are necessary to visualize their respective distributions clearly.

Comparison with State-of-the-Art Catalysts: The manuscript should include an extended discussion comparing the performance of the Ni-Zn nanocrystals with state-of-the-art catalysts for both acetylene hydrogenation in ethylene feedstocks and Z-selective semihydrogenation of alkynes in fine chemical synthesis. A detailed comparison regarding activity, selectivity, and tolerance towards functional groups will provide valuable insights into potential applications of the nanocrystal catalysts and their advantages over existing technologies.

Mechanistic Aspects and Recent Literature: The manuscript includes calculated mechanistic aspects (e.g., SI, Fig. 22) that would benefit from being compared to recent literature confirming catalysis on Ni₃ triangular sites, such as the publication in *Angew. Chem. Int. Ed.* 2023, 62, e202308790.

In summary, while the reported science is fundamentally interesting, the manuscript requires significant revisions to address the issues mentioned above. These include a thorough discussion of contradictory findings with key literature (especially Ref. 27), extension of the

investigation to more Zn-rich phases (e.g. NiZn₃), comprehensive characterization of the nanocrystals before and after catalysis and sufficient evidence excluding surface oxidation.

Recommendation: Reconsideration after major revision.”

- We are content to read that the reviewer finds the manuscript interesting and compelling and list our responses to the remarks and questions from reviewer 3 below.
- Here, there seems to be a misunderstanding. We believe our findings are in line with the work from F. Studt et al.⁵ The work from Studt et al. indicates that NiZn in a 1:1 stoichiometry is the most interesting alloy composition. We find experimentally that our NCs with a Ni₃Zn composition provide the best activity and selectivity. These NCs in fact have an approximate 1:1 surface composition (i.e. NiZn). This was described previously in the SI, but we now clarify this in the main text.
- The reviewer points out that surface oxidation might reduce the catalytic activity of the NCs. We indeed noticed the harmful effect of oxidation and therefore, we took much care to keep all NCs in a pristine, non-oxidized state. We provide below a Ni 2p_{3/2} XPS spectrum of air-exposed Ni NCs, which were partially oxidized on purpose. Compared to the XPS-spectra reported for the pristine as-synthesized NCs in the manuscript, these air exposed NCs feature 2 peaks, attributed to Ni(0) and Ni(II) species, unlike the pristine non-oxidized NCs reported in the manuscript, which feature a single peak attributed to Ni(0) species, Supplementary Fig. 6. We also provide below a Zn 2p_{3/2} XPS spectrum of Ni-Zn NCs, which were oxidized on purpose. Here, an O KLL peak is clearly observable, while absent in the pristine non-oxidized NCs reported in the manuscript, Supplementary Fig. 7.

Ni 2p 3/2 [eV]	species
852.3	Ni(0)
855.4	Ni(II)
861.5	Ni satellite

In addition to the XPS measurements shown for the NCs in Supplementary Figs. 6-7, the zerovalency is also observable from the acquired XANES measurements shown in Supplementary Fig. 5.

- We now provide more and higher quality Energy X-ray Dispersive Spectroscopy elemental mappings in Fig. S16.
- We thank the reviewer for suggesting that providing a better overview of and comparison to the semihydrogenation literature would benefit the manuscript. We have expanded the introduction considering the various catalyst designs existing in the literature for alkyne semihydrogenation in both homogeneous and heterogeneous catalysis fields. Note that directly comparing the semihydrogenation catalysis performance of the reported catalysts to the literature is tricky. Since we performed liquid-phase hydrogenation catalysis experiments in batch, it is difficult to compare our materials to the semihydrogenation catalysts studied for purifying acetylene gas streams in the literature. In addition, many literature reports on liquid-phase hydrogenation catalysis experiments in batch neglect to account for carbon balances. Carbon balances are important as they are a check for the occurrence of oligomerization and cyclotrimerization side-reactions. We nonetheless think that the reviewer raises an interesting point, and we therefore introduced a table in the SI, (Supplementary Table 10), which lists the performance of the presented Ni₃Zn catalyst in comparison to literature examples. For this table, we considered the literature reporting the semihydrogenation of 1-phenyl-1-propyne, as internal alkynes are less prone to side reactions such as oligomerization during hydrogenation reactions, compared to external

alkynes such as 1-phenylacetylene (for which literature results are hard to trust when carbon balances are not reported due to the likeness of side-reactions, such as oligomerization, occurring).

- We have added a small discussion and references to publications discussing semihydrogenation catalysis on Ni₃ triangular sites in the section discussing the DFT-calculations in the manuscript, as suggested by the reviewer.^{6,7} However, we point out that the catalyst materials (polyhydride Ni-Ga clusters and Ni₅Ga₃ intermetallic NCs) in papers discussing Ni₃ as an active site for selective semihydrogenation catalysis are structurally quite different from the materials in our manuscript (fcc-structured solid solution alloy NCs of Ni with Zn, Ga and In). In the case of our materials, we find that a parallelogram site involving 3 Ni-atoms and 1 atom of an alloying element to be more favorable for adsorbing alkynes compared to a Ni₃ triangular site, as shown in Supplementary Table 8 and Supplementary Figs. 21-22.

Reviewer 4.

“Disclaimer up front: This is a referee from the field of homogeneous catalysis, I therefore cannot comment on the preparation and characterization of the nanocrystals used for the catalytic experiments. In a similar vein, the reader, in the eyes of this referee, could be guided a bit better with respect to which types of catalysts generally perform well from the homogeneous as well as the heterogeneous field. Next to the precious metal catalysts, there have been plenty of 3d elements that can be employed as catalytically active center.

The present paper describes the use of Ni/Zn nanocrystals (which have been prepared by an amalgamation seeded growth technique) and their use in catalytic alkyne semihydrogenations. Generally, the quest in these transformations is to reach high selectivity for the alkene (and stereoselectivity) along with high catalytic performance at low H₂ pressure. Over the last decade, several steps ahead in terms of stereo- and chemoselectivity have been made especially with the use of 3d metals, but the pressure limitation is still challenging. In this light, the present work makes an important contribution, as the nanocrystals are catalytically active at 1 bar H₂ pressure. The authors claim that a variety of alkynes can be transformed to the corresponding alkenes. The high selectivity and reactivity is backed up by computational data.

Considering the substrate scope and the selectivities for the individual reactions: How are the stereo- and chemoselectivities determined? The authors claim that there has been GC analysis of all reactions, but there is no data supporting this in the SI. While for the standard reactions with phenylacetylene, the corresponding Z and E alkenes are commercially available and can be used as standards, this is not the case for most of the substrates shown. One can also not deduce that one particular stereoisomer will always appear first in the chromatogram. Therefore, data is needed to support the claims made in the paper.

Furthermore, many of the substrates carry functional groups that also may be reduced, such as ketones, aldehydes, nitro and ester groups. Can overreductions be excluded? How has this been determined? Have the authors observed any protodehalogenation with the bromides?

In Fig 5a and Fig 6 the authors speak of selectivity for the Z-alkene for most of the substrates. However, the large majority here are terminal alkynes which cannot deliver E or Z products. Therefore, the selectivity indication makes no sense. (Or have there been reactions with D2 to determine the E / Z selectivity?)

Catalytic semihydrogenations of 1,3-diynes are rare (see the works by Kirchner and Teichert, for example), and the stereoselectivity would be of interest here. Which stereoisomers are formed with the nanocrystals?

As far as this referee can see, no substrate has been isolated and the yield has been determined. Can the authors exclude a typical cyclotrimerization of alkynes as side-reaction? This is a commonly observed process reducing the overall yield.

Overall, this referee is of the opinion that the new type of catalyst described is definitely of high interest for the community. Before publication, the data to back up the claims in the present manuscript should be given.”

- We have taken the advice of the reviewer and expanded the introduction section to better inform the reader on the design approaches existing for homogeneous and heterogeneous alkyne semihydrogenation catalysts in the literature, adding: “For homogeneous catalysis, this includes non-precious metal complexes, e.g. Cu, Ni, Fe and Mn, including N-heterocyclic carbene (NHC),⁸⁻¹² pincer,¹³⁻¹⁸ bipyridine,¹⁹ metalloligand^{6,20} and phosphine²¹⁻²⁴ complexes.” as well as adding more references for heterogeneous alkyne semihydrogenation catalysts.
- All reaction solutions were analyzed using Gas Chromatography Mass Spectroscopy (GC-MS) after all experiments. The percentages of observed reactants and products after the experiment were obtained after integrating and normalizing the peaks obtained from the GC-MS analysis. The data values reported in Figures 2, 4-6, Supplementary Fig. 9 and Supplementary Tables 2-4, 9, 12-13 are derived from GC-MS analysis, and GC coupled with MS allowed the determination of the stereochemistry of the corresponding alkenes.
- The overreduction of substrates carrying reducible groups in addition to the alkyne functionality was checked for with GC-MS analysis after the experiments. When they occurred, e.g. in case of 4-nitro-phenylacetylene to a minor extent, an entry was made in Supplementary Tables 12-13 that report the data derived from the GC-MS analysis after the experiments. Mostly, no overreductions were discovered to take place, indicating that the Ni₃Zn NCs are chemoselective catalysts.
- In the substrate scope experiments, two alkyne substrates with a bromo-functionality, 4-Br-phenyl-phenyl-acetylene and 4,4'-Br-phenyl-phenyl-acetylene were tested (respectively substrates **27** and **28** reported in the manuscript in Fig. 6 and Supplementary Table 13). 0% conversion was observed for 4-Br-phenyl-phenyl-acetylene, making any assessment of the products impossible. In case of substrate **28**, 73 % conversion did occur under the employed conditions, yielding the corresponding

alkene with 96 % selectivity. The overhydrogenated product (to a minor extent, 4 %) was the only other product found after GC-MS analysis. No protodehalogenated products were found.

- Fig. 5a and Fig 6 include terminal and internal alkynes, of which the latter can be converted into cis-trans isomers of alkenes. The selectivity values are indicated for the Z-isomer, if the stereoisomer can be formed. If not (in case of terminal alkenes), the value indicates the selective formation of the alkene from semihydrogenation of the alkyne. We have adjusted the captions of Fig. 5a and Fig. 6 to explain the selectivity value more clearly.
- We appreciate that the reviewer points out the challenge of catalytically semihydrogenating some of the alkynes chosen for the substrate scope in the manuscript (we did not restrict ourselves to testing examples of substrates which are frequently reported to be easily semihydrogenated). The stereoselectivity values for this substrate are reported in Supplementary Table 13.
- This question on if cyclotrimerization of alkynes as a side-reaction can be excluded goes back to the point raised about the GC analysis and the new text describing our analysis should clarify this. Specifically, side-products formed during the semihydrogenation reaction are checked for with GC-MS analysis of the reaction mixtures after the experiments. While we did not observe any cyclotrimerization products in the GC-MS analysis, we cannot exclude that cyclotrimerization reactions, as well as oligomerization reactions, occurred and yielded products of very large molecular masses, which did not elute from the GC column during the analysis period of the GC-MS experiment. We have therefore calculated the carbon balance of each experiment. Concretely, this means that we calculated the total amount of carbon atoms of the remaining reactant, and the products formed after the reaction with GC-MS and compare this to the initial reactant substrate (the solution of which was also analyzed with GC-MS). When the carbon balance was not 100 %, we attributed this to side-reactions such as cyclotrimerization or oligomerization reactions (see for instance Figure 2). Because we measured the initial solution containing the reactant as well with GC-MS, we always obtained exact yields for the formed products. By taking carbon balances into account to check for potential non GC-detectable side-products due to reactions such as cyclotrimerization or oligomerization, we went further in our product analysis than most literature reports on semihydrogenation catalysis.
- We thank the reviewer for the very favorable opinion of our work and the useful suggestions provided to improve our manuscript. We hope that the explanation of how the GC-MS analysis was executed and where it is reported in the manuscript clarifies this.

References

- 1 Lai, X. *et al.* Critical review of life cycle assessment of lithium-ion batteries for electric vehicles: A lifespan perspective. *Etransportation* **12**, doi:ARTN 10016910.1016/j.etrans.2022.100169 (2022).
- 2 Akl, D. F. *et al.* Reaction-Induced Formation of Stable Mononuclear Cu(I)Cl Species on Carbon for Low-Footprint Vinyl Chloride Production. *Adv. Mater.* **35**, doi:10.1002/adma.202211464 (2023).
- 3 Huijbregts, M. A. J. *et al.* ReCiPe2016: a harmonised life cycle impact assessment method at midpoint and endpoint level. *Int. J. Life Cycle Ass.* **22**, 138-147, doi:10.1007/s11367-016-1246-y (2017).
- 4 Holder, C. F. & Schaak, R. E. Tutorial on Powder X-ray Diffraction for Characterizing Nanoscale Materials. *Acs Nano* **13**, 7359-7365, doi:10.1021/acsnano.9b05157 (2019).
- 5 Studt, F. *et al.* Identification of non-precious metal alloy catalysts for selective hydrogenation of acetylene. *Science* **320**, 1320-1322, doi:10.1126/science.1156660 (2008).
- 6 Muhr, M. *et al.* Catalytic Alkyne Semihydrogenation with Polyhydride Ni/Ga Clusters. *Angew. Chem. Int. Edit.* **62**, doi:10.1002/anie.202308790 (2023).
- 7 Cao, Y. Q. *et al.* Adsorption Site Regulation to Guide Atomic Design of Ni-Ga Catalysts for Acetylene Semi-Hydrogenation. *Angew. Chem. Int. Edit.* **59**, 11647-11652, doi:10.1002/anie.202004966 (2020).
- 8 Wakamatsu, T., Nagao, K., Ohmiya, H. & Sawamura, M. Copper-Catalyzed Semihydrogenation of Internal Alkynes with Molecular Hydrogen. *Organometallics* **35**, 1354-1357, doi:10.1021/acs.organomet.6b00126 (2016).
- 9 Thiel, N. O. & Teichert, J. F. Stereoselective alkyne semihydrogenations with an air-stable copper(I) catalyst. *Org. Biomol. Chem.* **14**, 10660-10666, doi:10.1039/c6ob02271e (2016).
- 10 Kaicharla, T., Zimmermann, B. M., Oestreich, M. & Teichert, J. F. Using alcohols as simple H₂-equivalents for copper-catalysed transfer semihydrogenations of alkynes. *Chem. Commun.* **55**, 13410-13413, doi:10.1039/c9cc06637c (2019).
- 11 Thiel, N. O., Kemper, S. & Teichert, J. F. Copper(I)-catalyzed stereoselective hydrogenation of 1,3-diynes and enynes. *Tetrahedron* **73**, 5023-5028, doi:10.1016/j.tet.2017.05.029 (2017).
- 12 Pape, F., Thiel, N. O. & Teichert, J. F. Z-Selective Copper(I)-Catalyzed Alkyne Semihydrogenation with Tethered Cu-Alkoxide Complexes. *Chem-Eur. J.* **21**, 15934-15938, doi:10.1002/chem.201501739 (2015).
- 13 Hale, D. J., Ferguson, M. J. & Turculet, L. (PSiP)Ni-Catalyzed (E)-Selective Semihydrogenation of Alkynes with Molecular Hydrogen. *Acs Catal* **12**, 146-155, doi:10.1021/acscatal.1c04537 (2022).
- 14 Garbe, M. *et al.* Chemoselective semihydrogenation of alkynes catalyzed by manganese(i)-PNP pincer complexes. *Catal. Sci. Technol.* **10**, 3994-4001, doi:10.1039/d0cy00992j (2020).
- 15 Pandey, D. K., Khaskin, E., Pal, S., Fayzullin, R. R. & Khusnutdinova, J. R. Efficient Fe-Catalyzed Terminal Alkyne Semihydrogenation by H₂: Selectivity Control via a Bulky PNP Pincer Ligand. *Acs Catal.* **13**, 375-381, doi:10.1021/acscatal.2c04274 (2023).
- 16 Srimani, D., Diskin-Posner, Y., Ben-David, Y. & Milstein, D. Iron Pincer Complex Catalyzed, Environmentally Benign, E-Selective Semi-Hydrogenation of Alkynes. *Angew. Chem. Int. Edit.* **52**, 14131-14134, doi:10.1002/anie.201306629 (2013).
- 17 Gorgas, N. *et al.* Efficient Z-Selective Semihydrogenation of Internal Alkynes Catalyzed by Cationic Iron (II) Hydride Complexes. *J. Am. Chem. Soc.* **141**, 17452-17458, doi:10.1021/jacs.9b09907 (2019).

- 18 Chowdhury, D., Goswami, S., Krishna, G. R. & Mukherjee, A. Transfer semi-hydrogenation of terminal alkynes with a well-defined iron complex. *Dalton T.* **53**, 3484-3489, doi:10.1039/d3dt03248e (2024).
- 19 Lee, M. Y., Kahl, C., Kaeffer, N. & Leitner, W. Electrocatalytic Semihydrogenation of Alkynes with [Ni(bpy)]. *Jacs Au* **2**, 573-578, doi:10.1021/jacsau.1c00574 (2022).
- 20 Ramirez, B. L. & Lu, C. C. Rare-Earth Supported Nickel Catalysts for Alkyne Semihydrogenation: Chemo- and Regioselectivity Impacted by the Lewis Acidity and Size of the Support. *J. Am. Chem. Soc.* **142**, 5396-5407, doi:10.1021/jacs.0c00905 (2020).
- 21 Chen, T. Q., Xiao, J., Zhou, Y. B., Yin, S. F. & Han, L. B. Nickel-catalyzed (*E*)-selective semihydrogenation of internal alkynes with hypophosphorous acid. *J. Organomet. Chem.* **749**, 51-54, doi:10.1016/j.jorganchem.2013.09.023 (2014).
- 22 Thiel, N. O., Kaewmee, B., Ngoc, T. T. & Teichert, J. F. A Simple Nickel Catalyst Enabling an *E*-Selective Alkyne Semihydrogenation. *Chem-Eur. J.* **26**, 1597-1603, doi:10.1002/chem.201903850 (2020).
- 23 Barrios-Francisco, R. & García, J. J. Semihydrogenation of alkynes in the presence of Ni(0) catalyst using ammonia-borane and sodium borohydride as hydrogen sources. *Appl. Catal. a-Gen.* **385**, 108-113, doi:10.1016/j.apcata.2010.06.052 (2010).
- 24 Thiel, N. O., Kaewmee, B., Ngoc, T. T. & Teichert, J. F. A Simple Nickel Catalyst Enabling an *E*-Selective Alkyne Semihydrogenation. *Chem-Eur. J.* **26**, 1597-1603, doi:10.1002/chem.201903850 (2020).

Response to Reviewers

Reviewer 3.

“I would like to begin by acknowledging the authors for addressing the majority of my comments in a satisfactory manner. However, I still have two comments, primarily regarding the explanations related to the surface composition of the nanocrystals.

First, I appreciate the authors response to my initial concern about the catalytic performance of Ni₃Zn being seemingly in contradiction to the literature. They propose that the surface of the Ni₃Zn nanocrystals has a near NiZn stoichiometry, supported by XPS data. Indeed, there is strong alignment between both the Ni(2p_{3/2}) and Zn(2p_{3/2}) peaks for Ni₃Zn and NiZn NCs, which suggests a similar surface composition. The STEM/EDX images (SI Fig. 16) show an even distribution of Ni and Zn over a larger area, to my opinion they do not provide detailed insight into the elemental distribution within individual particles due insufficient resolution (at least as depicted in SI Fig. 16). I would like to note that it appears to me there is an error in this figure, as the images for Ni₃Zn and NiZn display what seems to be the same object.

Second, since the surface compositions of Ni₃Zn and NiZn NCs are similar, it is noteworthy that their catalytic performance differs so significantly. A short discussion about that should be added to the main text.

In summary, I find the manuscript to be much improved after revision, and I appreciate the authors' efforts in addressing most of my concerns. The surface characterization of the Ni₃Zn NCs clarifies the apparent contradiction with the literature. I recommend accepting the manuscript for publication after these minor revisions.”

- We thank the reviewer for pointing out the error in Supplementary Fig. 16, where the same object was erroneously displayed for Ni₃Zn and NiZn NCs. We now display the correct STEM-EDX image for NiZn NCs in Supplementary Fig. 16 (which is now Supplementary Fig. 17 in the revised Supplementary Information). Due to the small size of the nanocrystals (below 4 nm) and the presence of organic ligands on their surface (which results in contamination under the focused electron beam when performing electron microscopy measurements), we were not able to obtain higher resolution STEM-EDX images.
- We are content to read that the reviewer appreciates our explanations considering the surface composition of the NCs. The reviewer raises the concern that Ni₃Zn and NiZn NCs are close in surface composition yet differing in activity. We find Ni₃Zn and NiZn NCs to be close in performance for alkyne semihydrogenation catalysis. They are similarly selective for the conversion of 1-phenyl-1-propyne and under mild reaction conditions (1 bar H₂, 80 °C, 10 h reaction time, 0.5 mol% catalyst) Ni₃Zn NCs display full conversion whereas NiZn NCs display 43% conversion. If harsher conditions were employed (which are normally required for catalysts based on non-precious metals) both Ni₃Zn NCs and NiZn NCs would show full conversion within a more similar timeframe. To clarify this in the manuscript, we added the sentence: **this decrease in reactivity for**

NiZn NCs compared to Ni₃Zn NCs is relatively small considering that the employed hydrogenation conditions, e.g., 1 bar H₂, are mild.

Reviewer 4.

“In the current version, the authors submit a significantly improved and re-worked version of the manuscript. Most of my comments to the previous version of the paper have been properly addressed. The key problem remains the claim of the selectivities (stereo- and chemoselectivity) of the alkyne semihydrogenation process: the authors describe at length that their analysis of yield and selectivity has been carried out by GC-MS analysis. While gas chromatography alone with some detectors is a quantitative technique when internal standards and calibration curves are used, GC-MS is not. An important reason for this is the non-linear behaviour of the molecules during ionization. Therefore, no precise analysis of the abovementioned parameters can be drawn from GC-MS.

As GCMS has been wrongly used as key method and the authors still do not show any chromatograms or spectra, the important claims of yields and selectivities - crucial descriptors for new and competitive catalysts - are not substantiated by proper data. Therefore, this referee suggests the rejection of the manuscript in its present form.”

- We understand the concern raised by reviewer 4, which we believe originates from incomplete information provided by us in the first round of revisions. Instead of only GC-MS, we employed Gas Chromatography coupled simultaneously to a Flame Ionization Detector and Mass Spectrometer GC-FID-MS, where the analyte is split both into MS and FID, which allowed us to use calibration curves and quantify carbon balances, thus resulting in quantitative yield and selectivity values.^{1,2} We apologize for our previous incomplete reply to the reviewer and to clarify the employed methodology better in the main text of the manuscript, we added: **To assess the NCs as catalysts (Fig. 2), we perform Gas Chromatography coupled simultaneously to a Flame Ionization Detector and Mass Spectrometer (GC-FID-MS), using *n*-tridecane as an internal standard, after each experiment.** In addition, we added a Figure (Supplementary Fig. 10) to the Supplementary Information where we provide an example case of the product analysis resulting from GC-FID-MS spectra for the reaction products obtained from the hydrogenation of 1-phenyl-1-propyne catalyzed by Ni₃Zn NCs.

Note that the method section in the manuscript contains a full description of the employed GC methodology, as copied below:

Gas Chromatography. GC measurements were performed on a Shimadzu-QP 2010 Ultra using an HP-5 column (30 m x 250 μ m x 0.25 μ m), with a column flow of 4.7 mL/min using helium as carrier gas, a FID (40:400 H₂/Air, 30 mL/min He makeup flow) and a MS operating with scan rate of 0.1 s from 50 to 350 m/z. Sample injection was done by a Shimadzu AOC-5000 Plus autosampler, 1 μ L in split mode (split ratio 30:1) with an injection port temperature of 310 °C. The longest method used for the analysis of the heaviest substrates consisted of the following temperature program: 4 min at 35 °C, then a ramp of 5 °C/min to 45 °C, hold for 3

min, ramp of 25 °C/min to 95 °C, hold for 1 min, ramp of 5 °C/min to 275 °C and hold for 12 min (total 60 min). Aliquots (0.1 mL) of reaction solutions were diluted with toluene (0.4 mL).

GC quantification was performed with the FID chromatogram integrated peak areas calibrated against each MS-detected compound's response factor relative to *n*-tridecane internal standard, under the assumption that response factors for the respective alkynes, alkenes and alkanes are the same. The quantification gave amounts of formed products, as well as remaining substrate that were used to calculate reported values of conversion (X = conversion) and selectivity (S = selectivity).

$$X_{\text{Substrate}} = \frac{n_{\text{Substrate}}^{\text{Initial}} - n_{\text{Substrate}}}{n_{\text{Substrate}}^{\text{Initial}}} * 100 \% \quad (3)$$

$$S_{\text{Semihydrogenation}} = \frac{n_{\text{All alkenes}}}{n_{\text{Total products}}} \quad (4)$$

$$S_{\text{Alkene}} = \frac{n_{\text{cis-Alkenes}}}{n_{\text{Total products}}} \quad (5)$$

$$\text{Carbon Balance} = \frac{n_{\text{Substrate}}^{\text{Initial}}}{n_{\text{Total products}} + n_{\text{Substrate}}} \quad (6)$$

The amount of total products formed is the sum of MS-detected products, including *cis/trans*-alkenes, alkene chain isomers dimers/trimers and alkanes. In general, the carbon balance closes within a few percent (2-3%) of 100%. Carbon balances below 97% are attributed to formation of oligomers and/or strong binding of products or substrates to the catalyst that are not detectable via GC. The reported selectivity values are lower estimates, calculated under the assumption that oligomers are responsible for the <100% carbon balances and are hence considered as products.

References

- 1 Xiong, J. Q. *et al.* Mediating trade-off between activity and selectivity in alkynes semi-hydrogenation via a hydrophilic polar layer. *Nat. Commun.* **15**, doi:ARTN 122810.1038/s41467-024-45104-6 (2024).
- 2 Gao, Y. *et al.* Field-induced reagent concentration and sulfur adsorption enable efficient electrocatalytic semihydrogenation of alkynes. *Sci. Adv.* **8**, doi:ARTN eabm947710.1126/sciadv.abm9477 (2022).

Reviewer #4 (Remarks to the Author):

“The authors have now responded to the referees comments and have clarified that they have indeed used GC-FID-MS as a method to quantify their results and in this revision there is one single example in which they show how the data was analyzed and how the selectivities of the alkyne (semi)hydrogenation was obtained.

From my perspective, in order to assess the selectivity, purity and even identity of the 20+ substrates with quite some structural diversity and different functional groups, I would have expected characterization of the products. I would have expected at least the GC traces for the individual products, as there is no characterization of the products anyways.

I leave the decision to the editors as to whether the provided data is sufficient to back up the claims made in the paper and whether or not one should believe that the claims made in the paper are true without any data for the individual molecules prepared.”

The reviewer requests to provide product characterization for the entire substrate scope, for instance by providing the GC traces for the individual products which were used to obtain the conversion and selectivity values reported in the manuscript. Previously, we showed GC-FID-MS data for 1 substrate to illustrate our employed methodology. To address the request from the reviewer, we now provide the GC-FID-MS data, Supplementary Figs. 40-70, for all 31 alkynes investigated in the substrate scope, including chromatograms and the MS spectra of the substrates and formed reaction products together with their corresponding chemical structures.